# Effects of $^{238}$U variability and physical transport on water column $^{234}$Th downward fluxes in the coastal upwelling system off Peru

Ruifang C. Xie[1]*, Frédéric A. C. Le Moigne[2], Insa Rapp[1], Jan Lüdke[1], Beat Gasser[3], Marcus Dengler[1], Volker Liebetrau[1], Eric P. Achterberg[1]

[1]GEOMAR Helmholtz Center for Ocean Research Kiel, Wischhofstrasse 1-3, 24148 Kiel, Germany

[2]Mediterranean Institute of Oceanography (UM 110, MIO), CNRS, IRD, Aix Marseille Université, Marseille, France

[3]IAEA Environment Laboratories, 4 Quai Antoine 1er, 98000 Monaco Monaco

* corresponding author: rxie@geomar.de

**Abstract**

The eastern boundary region of the southeastern Pacific Ocean hosts one of the world's most dynamic and productive upwelling systems with an associated oxygen minimum zone (OMZ). The variability in downward export fluxes in this region, with strongly varying surface productivity, upwelling intensities and water column oxygen content, is however poorly understood. Thorium-234 ($^{234}$Th) is a powerful tracer to study the dynamics of export fluxes of carbon and other elements, yet intense advection and diffusion in nearshore environments impact the assessment of depth-integrated $^{234}$Th fluxes when not properly evaluated. Here we use VmADCP current velocities, satellite wind speed and *in situ* microstructure measurements to determine the magnitude of advective and diffusive fluxes over the entire $^{234}$Th flux budget at 25 stations from 11°S to 16°S in the Peruvian OMZ. Contrary to findings along the GEOTRACES P16 eastern section, our results showed that weak surface wind speed during our cruises induced low upwelling rates and minimal upwelled $^{234}$Th fluxes, whereas vertical diffusive $^{234}$Th fluxes were important only at a few shallow shelf stations. Horizontal advective and diffusive $^{234}$Th fluxes were negligible because of small alongshore $^{234}$Th gradients. Our data indicated a poor correlation between seawater $^{238}$U activity and salinity. Assuming a linear relationship between the two would lead to significant underestimations of the total $^{234}$Th flux by up to 40% in our study. Proper evaluation of both physical transport and variability in $^{238}$U activity is thus crucial in coastal $^{234}$Th flux studies. Finally, we showed large temporal variations on $^{234}$Th residence times across the Peruvian upwelling zone, and cautioned future carbon export studies to take these temporal variabilities into consideration while evaluating carbon export efficiency.

**Keywords:** eastern tropical South Pacific, $^{234}$Th tracer, uranium-salinity correlation, physical processes, residence time

## 1. Introduction

Isotopes of thorium (Th) are widely used as tracers for particle cycling in the oceans (Waples et al., 2006). In particular, $^{234}$Th has been extensively used to trace particle dynamics and export fluxes in the upper ocean, and to quantify the marine budgets of important macro- and micronutrients such as carbon (C), nitrogen (N), phosphorus (P) and iron (Fe) (e.g. Bhat et al., 1968; Buesseler et al., 1992; Coale and Bruland, 1987; Lee et al., 1998; Le Moigne et al., 2013; Cochran and Masqué, 2003; Van Der Loeff et al., 2006; Black et al., 2019). $^{234}$Th has a relatively short half-life ($\tau_{1/2} = 24.1$ days) that allows studies of biological and physical processes occurring on timescales of days to weeks. Unlike its radioactive parent uranium-238 ($^{238}$U, $\tau_{1/2} = 4.47$ Ga) that is soluble in seawater, $^{234}$Th is highly particle reactive with a particle-water partition coefficient of $10^3$ to $10^8$ (Santschi et al., 2006 and references therein) and is thus strongly scavenged by particles (Bhat et al., 1968). Generally, a deficit of $^{234}$Th relative to $^{238}$U is observed in the surface ocean and reflects net removal of $^{234}$Th due to particle sinking, whereas secular equilibrium between $^{234}$Th and $^{238}$U is observed for intermediate and deep waters. Integrating this surface $^{234}$Th deficit with depth yields the sinking flux of $^{234}$Th and, if elemental:$^{234}$Th ratios are known, the sinking flux of elements such as C, N, P, Si and trace metals (e.g. Bhat et al., 1968; Buesseler et al., 1998; Buesseler et al., 1992; Coale and Bruland, 1987; Weinstein and Moran, 2005; Buesseler et al., 2006; Owens et al., 2015; Black et al., 2019; Puigcorbé et al., 2020).

Various $^{234}$Th models have been put forward to study adsorption/desorption, aggregation and export, but single box models that assume negligible $^{234}$Th fluxes due to physical transport are commonly used to calculate oceanic $^{234}$Th-derived particle fluxes (see detailed review by Savoye et al., 2006). This assumption is typically appropriate in open ocean settings where $^{234}$Th fluxes due to advection and diffusion are small relative to the downward fluxes of $^{234}$Th associated with particle sinking. However, in upwelling regions

such as the equatorial Pacific and coastal systems, advective and diffusive [234]Th fluxes may
become increasingly important (e.g., Bacon et al., 1996; Buesseler et al., 1998; Buesseler et
al., 1995; Dunne and Murray, 1999). For example, in the equatorial Pacific, strong upwelling
post El-Niño could account for ~50% of the total [234]Th fluxes (Bacon et al., 1996; Buesseler
et al., 1995). Ignoring the upwelling term could thus lead to an underestimation of [234]Th
fluxes by a factor of 2. Conversely, horizontal diffusion carrying recently upwelled, [234]Th-
replete waters has been shown to balance the upwelled [234]Th fluxes in the central equatorial
Pacific (Dunne and Murray, 1999). To the contrary, advective and diffusive [234]Th fluxes were
minimal off the Crozet Islands in the Southern Ocean due to limited horizontal [234]Th
gradients, long residence time of water masses, and low upwelling rates and diffusivities
(Morris et al., 2007).

The dynamic nature of coastal processes requires that physical terms should be

included in [234]Th flux calculation whenever possible. Accurate measurements of current
velocities and diffusivities are however challenging and thus direct observations of the effects
of physical processes on [234]Th distributions in coastal regions are scarce. Limited studies have
incorporated advection and diffusion in the nearshore zones of the Arabian Sea (Buesseler et
al., 1998), Gulf of Maine (Gustafsson et al., 1998; Benitez-Nelson et al., 2000), the South
China Sea (Cai et al., 2008) and Peruvian oxygen minimum zone (OMZ) (Black et al., 2018).
In the Arabian Sea, coastal upwelling during the southwest monsoon season could account for
over 50% of the total [234]Th flux (Buesseler et al., 1998). Horizontal advection has been shown
to be substantial in the Inner Cosco Bay of the Gulf of Maine (Gustafsson et al., 1998),
whereas offshore advection and diffusion are only important in late summer (Benitez-Nelson
et al., 2000). Therefore, the importance of physical processes on the [234]Th flux estimate is
highly dependent on the seasonal and spatial variability of the current velocities, diffusivities
and [234]Th gradients. In terms of the Peruvian OMZ, Black et al. (2018) showed that coastal
upwelling accounts for >50% of total $^{234}$Th fluxes at 12°S; however, how upwelling $^{234}$Th
fluxes vary seasonally and spatially in this region is unclear.
Another uncertainty in $^{234}$Th flux calculations in such region stems from variations on
dissolved $^{238}$U activities. Generally speaking, U behaves conservatively under open ocean
oxic conditions and is linearly correlated with salinity (Chen et al., 1986; Ku et al., 1977;
Owens et al., 2011). However, numerous studies have shown that such correlation breaks
down in various marine environments including the tropical Atlantic (Owens et al., 2011),
Mediterranean Sea (Schmidt and Reyss, 1991), and Arabian Sea (Rengarajan et al., 2003).
Although it is generally accepted that deviations from the linear $^{238}$U-S correlation will lead to
differences in the final calculated $^{234}$Th fluxes, there is currently little knowledge on how
significant these differences could be.
In this study, we report vertical profiles of $^{234}$Th and $^{238}$U along four transects
perpendicular to the coastline of Peru (i.e. shelf-offshore transects). We evaluate the $^{238}$U-S
correlation in low-oxygen waters and how deviations from this correlation impact final $^{234}$Th
flux estimates. We also assess the spatial and temporal importance of advection and diffusion
on $^{234}$Th flux estimates.

**2. Sampling and methods**

2.1 Seawater sampling and analysis
Seawater samples were collected at 25 stations along 4 shelf-offshore transects
between 11°S and 16°S in the Peruvian OMZ during two cruises M136 and M138 on board
the RV Meteor (Figure 1). Cruise M136 took place in austral autumn (April 11 to May 3,
2017) along two main transects at 12°S and 14°S (Dengler and Sommer, 2017). Two stations
from M136 (stations 458 and 495) were reoccupied within a week (repeat stations 508 and

516, respectively) to evaluate the steady-state assumption in the $^{234}$Th flux calculation. The

surface sample of the repeat station 508 (reoccupied 4.5 days after station 458) was missing so

only results from repeat stations 495 and 516 (occupation interval 1.5 days) were compared

and discussed in terms of the non-steady state model (section 3.3). $^{234}$Th sampling during

cruise M138 was carried out in austral winter (June 1 to July 4, 2017) and focused on four

shelf-offshore transects at 11°S, 12°S, 14°S and 16°S.

At each station, a stainless-steel rosette with Niskin bottles (Ocean Test Equipment®)

was deployed for sampling of total $^{234}$Th in unfiltered seawater and dissolved $^{238}$U (0.2 μm

pore size, Acropak® polycarbonate membrane). High vertical resolution sampling was

performed in the upper 200 m where most of the biological activity occurs; additional depths

were sampled down to 600 m, or 50 m above the seafloor. Deep seawater at 1000 m, 1500 m,

and 2000 m was sampled at three stations to determine the absolute β counting efficiency.

Salinity, temperature, oxygen concentrations and fluorescence data (Table S1) were derived

from the sensors (Seabird Electronics® 9plus system) mounted on the CTD frame (Krahmann,

2018; Lüdke et al., in review 2020).

Sample collection and subsequent chemical processing and analysis for total $^{234}$Th

followed protocols by Pike et al. (2005) and SCOR Working Group RiO5 cookbook

(https://cmer.whoi.edu/). Briefly, a $^{230}$Th yield tracer (1 dpm) was added to each sample (4 L)

before Th was extracted with $MnO_2$ precipitates. Precipitates were filtered onto 25 mm quartz

microfiber filters (Whatman® QMA, 2.2 μm nominal pore size) and dried overnight at 50℃,

after which they were counted at sea on a Risø® low-level beta GM multicounter until

uncertainty was below 3%, and again 6 months later at home laboratory for background $^{234}$Th

activities. After the second beta counting, filters were digested in an 8M $HNO_3$/10% $H_2O_2$

solution (Carl Roth®, trace metal grade). 10 dpm of $^{229}$Th was added to each sample at the

beginning of digestion to achieve a 1:1 atom ratio between $^{229}$Th:$^{230}$Th. Digested samples

were diluted in a 2.5% $HNO_3$/0.01% HF mixture and $^{229}$Th/$^{230}$Th ratios were measured using

an ICP-MS (ThermoFisher® Element XR) to determine the chemistry yield and final $^{234}$Th
activities. The average yield was calculated to be 97% ± 6% (n = 247). For a subset of
samples (marked in Table S1) whose analysis failed during initial ICP-MS measurement,
anion chromatography (Biorad® AG1x8, 100 – 200 mesh, Poly-Prep columns) was performed
to remove Mn from the sample matrix before another ICP-MS analysis. This subset of
samples also included three samples (marked in Table S1) whose initial ICP-MS measurement
was successful, to test whether anion chromatography affects final ICP-MS results. Identical
$^{229}$Th/$^{230}$Th ratios were measured for samples with and without column chromatography (see
Table S1 footnotes for details).
Each $^{238}$U sample was acidified to pH ~1.6 at sea and transported home for analysis.
Samples of dissolved $^{238}$U were diluted 20 times in 1N HNO$_3$ at home laboratory and spiked
with an appropriate amount of $^{236}$U spike to achieve $^{236}$U:$^{238}$U ~ 1:1. Ratios of $^{236}$U:$^{238}$U were
analyzed by ICP-MS (ThermoFisher Element XR) and activities of $^{238}$U were calculated using
isotope dilution. Seawater certified reference materials (CRMs), CASS-6 and NASS-7, and
the International Association for the Physical Sciences of the Oceans (IAPSO) standard
seawater were analyzed routinely for uranium concentrations.

2.2 Flux calculation
Assuming a one box model, the temporal change of $^{234}$Th activities is balanced by
production from $^{238}$U, radioactive decay of $^{234}$Th, removal of $^{234}$Th onto sinking particles, and
transport into or out of the box by advection and diffusion (Bhat et al., 1968; Savoye et al.,
2006; and references therein):
$$\frac{\partial A_{Th}}{\partial t} = \lambda(A_U - A_{Th}) - P + V \tag{1}$$

where $A_U$ and $A_{Th}$ are respectively the activities of dissolved $^{238}$U and total $^{234}$Th, $\lambda$ is
the decay constant of $^{234}$Th, P is the net removal flux of $^{234}$Th, and V is the sum of advective
and diffusive fluxes. It is recommended that the time interval between station occupations
should be >2 weeks in order to adequately capture the temporal variability of the mean spatial
gradients rather than small local changes (Resplandy et al., 2012). The solution of Eq. (1)
(Savoye et al., 2006) is
$$P = \lambda \left[ \frac{A_U(1-e^{-\lambda \Delta t}) + A_{Th1} \cdot e^{-\lambda \Delta t} - A_{Th2}}{1-e^{-\lambda \Delta t}} \right] \qquad (2)$$

where $\Delta t$ is the time interval between repeat occupations of a station; $A_{Th1}$ and $A_{Th2}$
are respectively total $^{234}$Th activities during the first and second occupation. At times when
repeat sampling is not possible within adequate cruise timeframe, steady state conditions are
generally assumed, i.e. $\frac{\partial A_{Th}}{\partial t} = 0$.  In this case, Eq. (1) is simplified into:
$$P = \int_0^z \lambda(A_U - A_{Th})dz + V \qquad (3)$$

The vertical flux of $^{234}$Th, P (dpm m$^{-2}$ d$^{-1}$), is integrated to the depth of interest. Earlier
studies generally used arbitrarily fixed depths (e.g., the base of mixed layer or ML, and 100
m) for $^{234}$Th and POC flux estimates (e.g., Bacon et al., 1996; Buesseler et al., 1992). Recent
studies emphasized the need to normalize POC flux to the depth of euphotic zone (EZ), which
separates the particle production layer in the surface from the flux attenuation layer below
(Black et al., 2018; Buesseler and Boyd, 2009; Rosengard et al., 2015). In the open ocean, the
depth of EZ is generally similar to ML depth. The PAR (Photosynthetically Active Radiation)
sensor was not available during both of our cruises, so that it was not possible to identify the
base of the EZ. For the purpose of this study, the slight difference of the exact depth chosen
(ML vs. EZ) was of little relevance to the significance of physical processes and $^{238}$U
variability. Due to sampling logistics, however, we did not sample at the base of the ML, but
5-20 m below the ML. This depth corresponded closely to the EZ depth used in Black et al.
(2018) in the same study area during austral spring 2013. For the purpose of comparison with
earlier studies which reported [234]Th fluxes at 100 m, we also calculated [234]Th fluxes at 100 m
in this study.

2.3 Quantification of the physical fluxes
The physical term V in Eq. (2) is expressed as following:
$$V = \int_0^z \left( w \frac{\partial Th}{\partial z} - u \frac{\partial Th}{\partial x} - v \frac{\partial Th}{\partial y} \right) dz + \int_0^z \left( K_x \frac{\partial^2 Th}{\partial x^2} + K_y \frac{\partial^2 Th}{\partial y^2} - K_z \frac{\partial^2 Th}{\partial z^2} \right) dz \qquad (3)$$
where $w$ is the vertical (i.e. upwelling) velocity (m s$^{-1}$), u and v respectively the zonal
and meridional current velocities (m s$^{-1}$), and $K_x$, $K_y$, and $K_z$ represent eddy diffusivities (m$^2$ s$^{-1}$
$^{1}$) in zonal, meridional and vertical directions, respectively. $\frac{\partial Th}{\partial z}$, $\frac{\partial Th}{\partial x}$ and $\frac{\partial Th}{\partial y}$ are vertical and
horizontal [234]Th gradients (dpm L$^{-1}$ m$^{-1}$), and $\frac{\partial^2 Th}{\partial x^2}$, $\frac{\partial^2 Th}{\partial y^2}$ and $\frac{\partial^2 Th}{\partial z^2}$ are respectively the second
derivative of [234]Th (dpm L$^{-1}$ m$^{-2}$) on the zonal, meridional and vertical directions.

2.3.1 Estimation of upwelling velocities
In the Mauritanian and Peruvian coastal upwelling regions, there is strong evidence
that upwelling velocities in the mixed layer derived from satellite scatterometer winds and
Ekman divergence (Gill, 1982) agree well with those from helium isotope disequilibrium
(Steinfeldt et al., 2015). The parameterization by Gill (1982) considers the baroclinic response
of winds blowing parallel to a coastline in a two-layer ocean. Vertical velocity (*w*) at the
interface yields
$$w = \frac{\tau}{\rho f a} e^{-x/a} \qquad (4)$$
where τ is the wind stress (kg m$^{-1}$ s$^{-2}$) parallel to the coast line, ρ the water density
(1023 kg m$^{-3}$), *f* the Coriolis parameter (s$^{-1}$) as a function of latitude, *a* the first baroclinic
Rossby radius (km) and *X* the distance (km) to the coast.
Upwelling velocities were calculated at stations within 60 nautical miles (nm) of the
coast, where upwelling is the most significant (Steinfeldt et al., 2015). We used *a* = 15 km for
all stations based on the results reported by Steinfeldt et al. (2015) for the same study area.
The magnitude of monthly wind stress was estimated from the monthly wind velocities
(Smith, 1988):
$$\tau = \rho_{air} C_D U^2 \tag{5}$$

where $\rho_{air}$ is the air density above the sea surface (1.225 kg m$^{-3}$), $C_D$ the drag
coefficient (10$^{-3}$ for wind speed < 6 m s$^{-1}$), and *U* the wind speed.
Monthly wind speed (m s$^{-1}$) fields from MetOp-A/ASCAT scatterometer sensor with
a spatial resolution of 0.25° (Bentamy and Croize-Fillon, 2010) were retrieved from the
Centre de Recherche et d'Exploitation Satellitaire (CERSAT), at IFREMER, Plouzané
(France) (data version numbers L3-MWF-GLO-20170903175636-01.0 and L3-MWF-GLO-
20170903194638-01.0). We assumed a linear decrease of *w* from base of the mixed layer
toward both the ocean surface and 240 m depth (bottom depth of our shallowest station).
Upwelling rates at any depth between 0 and 240 m at individual stations could thus be
determined once *w* was estimated. Following (Rapp et al., 2019), an error of 50% was
assigned to estimated upwelling velocities to account for uncertainties associated with the
spatial structure and temporal variability of the wind field, and the satellite wind product near
the coast.

2.3.2 Estimation of upper-ocean velocities

During both cruises a phased-array vessel-mounted acoustic Doppler current profiler (VmADCP; 75 kHz Ocean Surveyor, Teledyne RD-Instruments) continuously measured zonal and meridional velocities in the upper 700 m of the water column (Lüdke et al., in review 2020). Post-processing of the velocity data included water track calibration and bottom editing. After calibration, remaining uncertainty of hourly averages of horizontal velocities are smaller than 3 cm s$^{-1}$ (e.g. Fischer et al., 2003). For the horizontal advective flux calculation (Eq. 3), velocities collected within a 10 km radian at inshore stations (St. 353, 428, 458, 475, 508, 904, and 907) and within a 50 km radian at offshore stations (Lüdke et al., in review 2020) were averaged. Data collected at the same positions within 5 days due to station repeats were also included in the velocity average. As representative for the near-surface flow, we extracted the velocity data from the top 30 m for M136 stations and top 50 m for M138 station (defined as the "top layer" thereafter); these depths correspond to 5-20 m below the base of the ML during each cruise.

2.3.3 Estimation of vertical and horizontal eddy diffusivities

While the strength of ocean turbulence determines the magnitude of diapycnal or vertical eddy diffusivities, the intensity of meso- and submesoscale eddies determine the magnitude of lateral eddy diffusivities. During the R/V Meteor cruise M136 and the follow up cruise (M137) in the same region, the strength of upper-ocean turbulence was measured using shear probes mounted to a microstructure profiler. The loosely-tethered profiler was optimized to sink at a rate of 0.55 m s$^{-1}$ and equipped with three shear sensors, a fast-response temperature sensor, an acceleration sensor, two tilt sensors and conductivity, temperature, depth sensors sampling with a lower response time. On transit between each CTD station 3 to 9 microstructure profiles were collected. Standard processing procedures were used to determine the dissipation rate of turbulent kinetic energy ($\varepsilon$) in the water column (see

Schafstall et al., 2010 for detailed description). Subsequently, turbulent vertical diffusivities
$K_Z$ were determined from $K_z = \Gamma \varepsilon N^{-2}$ (Osborn, 1980), where $N$ is stratification and $\Gamma$ is the
mixing efficiency for which a value of 0.2 was used following Gregg et al. (2018).
Stratification (Buoyancy frequency) was calculated using CTD data retrieved from
microstructure profilers and following the gsw_Nsquared function from the Gibbs Sea Water
library (McDougall et al., 2009; Roquet et al., 2015). A running mean of 10 dbar was applied
to avoid including unstable events due to turbulent overturns. The 95% confidence intervals
for averaged $K_z$ values were determined from Gaussian error propagation following Schafstall
et al. (2010).

Altogether, 189 microstructure profiles were collected during M136 (Thomsen and

Lüdke, 2018) and 258 profiles during the follow-up cruise M137 (unpublished data; May 6 –
29, 2017). An average turbulent vertical diffusivity profile was calculated each from all
inshore (<500m water depth) and all offshore (>500m water depth) profiles (Figure S1).
Microstructure profiles collected during cruise M138 were not available but there were little
variations amongst the cruise average inshore and offshore microstructure profiles from M136
and M137 despite drastic change in the intensities of the poleward Peru Chile Undercurrent
(Lüdke et al., in review 2020). It thus appears appropriate to apply these average vertical
diffusivities also to stations during M138.

Horizontal eddy diffusivity could not be determined from data collected during the

cruises. Surface eddy diffusivities in the North Atlantic OMZ were estimated to be on the
order of a few 1000 $m^2$ $s^{-1}$ that decrease exponentially with depth (Hahn et al., 2014). Similar
magnitude of eddy diffusivities was estimated for the ETSP based on surface drifter data and
satellite altimetry (Abernathey and Marshall, 2013; Zhurbas and Oh, 2004). We thus consider
an eddy diffusivity of 1000 $m^2$ $s^{-1}$ as a good approximate in this study for the evaluation of
horizontal diffusive [234]Th fluxes.

2.4 Residence time of $^{234}$Th

The residence time ($\tau_{Th}$) of total $^{234}$Th represents a combination of the time required

for the partition of dissolved $^{234}$Th onto particulate matter and that for particle removal. In a
one-box model, the residence time of an element of interest can be estimated by determining
the standing stock of this element and the rates of elemental input to the ocean or the rate of
elemental removal from seawater to sediments (Bewers and Yeats, 1977; Zimmerman, 1976):

$$\tau_{Th} = \frac{A_{Th(mean)} \cdot Z}{P}$$                              (6)

For the case of $^{234}$Th, $A_{Th(mean)}$ is the averaged $^{234}$Th activities of the surface layer, $Z$ is

the depth of top layer, and P the removal flux of $^{234}$Th.

**3. Results**
3.1 Profiles of dissolved $^{238}$U, total $^{234}$Th, oxygen and fluorescence

The vertical profiles of $^{238}$U and $^{234}$Th activities are shown in Figure 2 and tabulated in

Table S1. Data from station 508 were reported in Figure 2 and Table S1 but excluded in the
Discussion section, because the surface sample at 5 m from this station was missing, which
prevents any flux calculation. Also tabulated in Table S1 are temperature, salinity and
concentrations of oxygen and fluorescence obtained from the CTD sensors. Uranium
concentrations of CRMs and the IAPSO standard seawater are reported in Table S2.

Activities of $^{238}$U showed small to negligible variations with depth, averaging 2.54 ±

0.05 dpm L$^{-1}$ (or 3.28 ± 0.07 ng/g, 1SD, n = 247) at all stations. The vertical distributions of
$^{238}$U did not appear to be affected by water column oxygen concentrations or the extent of
surface fluorescence maxima (Figure 2). Average U concentrations of both CASS-6 (2.77 ±
0.04 ng g$^{-1}$, 1SD, n = 5) and NASS-7 (2.86 ± 0.05 ng/g, 1SD, n = 5) measured in this study
agreed well with certified values (2.86 ± 0.42 ng g$^{-1}$ and 2.81 ± 0.16 ng g$^{-1}$, respectively).
Average $^{238}$U concentration measured in our IAPSO standard seawater (OSIL batch P156)
(3.24 ± 0.06 ng g$^{-1}$, 1SD, n = 27) is slightly higher than that reported in Owens et al. (2011)
(3.11 ± 0.03 ng g$^{-1}$, 1SD, n = 10, OSIL P149), and may reflect slight differences in U
concentrations between different OSIL batches.

Total $^{234}$Th activities varied from 0.63 to 2.89 dpm L$^{-1}$ (Figure 2). All stations showed

large $^{234}$Th deficits in surface waters with $^{234}$Th/$^{238}$U ratios as low as 0.25 (Figure 3). The
extent of surface $^{234}$Th deficits did not vary as a function of depths of either mixed layer or the
upper oxic-anoxic interface, nor the magnitude of surface fluorescence concentrations (Table
1, Figure 2). $^{234}$Th at all stations generally reached equilibrium with $^{238}$U at depths between 30
m and 250 m (Table 1). The equilibrium depths were slightly shallower toward the shelf at the
11°S, 12°S and 16°S transects. At St. 912, deficits of $^{234}$Th extended beyond 600 m depth
(Figure 2). The following stations (St. 428, 879, 898, 906, 907, 915, 919) displayed a
secondary $^{234}$Th deficit below the equilibrium depth, indicative of $^{234}$Th removal processes. A
small $^{234}$Th excess at depth was only observed for St. 559 at 100 m. Ratios of $^{234}$Th/$^{238}$U for
deep samples at 1000 m, 1500 m, and 2000 m varied between 0.95 and 1.02 (1.00 ± 0.04,
1SD, n = 11), suggesting that $^{234}$Th was at equilibrium with $^{238}$U at these depths.

3.2 Vertical and horizontal $^{234}$Th gradients

Discrete vertical $^{234}$Th gradients in each profile (or the curvature of the profile) were

estimated by the difference in $^{234}$Th activities and that in sampling depths. As such, vertical
$^{234}$Th gradients varied greatly amongst stations, and were larger at shallow depths ranging
from 0.003 dpm $L^{-1}$ $m^{-1}$ to 0.085 dpm $L^{-1}$ $m^{-1}$ (median 0.013 dpm $L^{-1}$ $m^{-1}$). Vertical $^{234}$Th
gradients were essentially negligible at and below equilibrium depths.

While calculation of the vertical $^{234}$Th gradient is straightforward, the same is hardly

true for the determination of horizontal $^{234}$Th gradient. Mean $^{234}$Th activities in the top layer
(see section 2.3.2 for depth definition) of the water column are highly variable amongst
stations (Table 3, Figure 4), and likely reflect variations occurring at small temporal and
spatial scales in the Peruvian OMZ. Quantification of the horizontal $^{234}$Th gradient between
individual station thus may not be adequate to evaluate large scale advection and eddy
diffusion across the study area. Therefore, alongshore $^{234}$Th gradients on a larger spatial scale
(1° apart) were instead calculated by grouping stations into 1° by 1° grids and averaging $^{234}$Th
activities of each grid for the top layer. Alongshore $^{234}$Th gradients in the top layer at
nearshore stations for M138 are fairly consistent, ranging from 1.5 x $10^{-6}$ dpm $L^{-1}$ $m^{-1}$ to 1.7 x
$10^{-6}$ dpm $L^{-1}$ $m^{-1}$, with a slightly stronger gradient in the north compared to the south. The net
difference in alongshore $^{234}$Th gradient is merely 2 x $10^{-7}$ dpm $L^{-1}$ $m^{-1}$. A slightly smaller
alongshore $^{234}$Th gradient of 4.8 x $10^{-7}$ dpm $L^{-1}$ $m^{-1}$ was observed for M136. The magnitude of
the net difference in alongshore $^{234}$Th gradient for M136 cannot be adequately quantified, due
to smaller spatial sampling coverage. Judging on the similarity in the spatial distributions of
mean $^{234}$Th between cruises M136 and M138 (Figure 4), it is reasonable to assume that net
difference in alongshore $^{234}$Th gradient remained similar during both cruises.

3.3 Steady state vs. non-steady state models

The relative importance of $^{234}$Th fluxes due to advection and diffusion were assessed

here assuming steady state conditions, which assume negligible temporal $^{234}$Th variability.
But how valid is this assumption in the Peruvian upwelling zone? Profiles of temperature and
oxygen at repeat stations 458 and 508 showed that a lightly cooler and oxygen-depleted water
mass dominated at the upper 50 m at station 508 (Figure 5). However, an assessment of the
$^{234}$Th fluxes at these two stations were not possible as the surface sample from station 508 was
missing. Repeat stations 495 and 516 show substantial temporal variations in $^{234}$Th activities
at each sampled depth in the top 200 m, while temperature and salinity profiles confirmed that
similar water masses were sampled during both occupations (Figure 5). Particularly, the
surface $^{234}$Th deficit was more intense at St. 495 ($^{234}$Th/$^{238}$U = 0.44) compared to St. 516
($^{234}$Th/$^{238}$U = 0.73). Correspondingly, $^{234}$Th fluxes decreased substantially from St. 495 to St.
516. At 100 m, the difference in $^{234}$Th fluxes between these two stations was ~ 30% (3200 ±
90 dpm m$^{-2}$ d$^{-1}$ at St. 495 and 2230 ± 110 dpm m$^{-2}$ d$^{-1}$ at St. 516). At 200 m where $^{234}$Th
resumed equilibrium with $^{238}$U at both stations, $^{234}$Th flux difference was ~ 25% (4510 ± 220
dpm m$^{-2}$ d$^{-1}$ at St. 495 and 3455 ± 200 dpm m$^{-2}$ d$^{-1}$ at St. 516). Taking the non-steady state
term in Eq. (1) into consideration (see details in Resplandy et al. (2012) and Savoye et al.
(2006) for the derivation of flux formulation and error propagation) increased total $^{234}$Th at St.
516 by 40% to 3110 ± 1870 dpm m$^{-2}$ d$^{-1}$ at 100 m (or 45% to 5040 ± 2290 dpm m$^{-2}$ d$^{-1}$ at 200
m), which is indistinguishable within error from fluxes at St. 495. The large errors associated
with the non-steady state calculation due to the short duration between station occupations
prevent a meaningful application of this model in the current study (also see discussion in
Resplandy et al, 2012). As estimation of the physical fluxes is independent of the models
chosen between steady and non-steady states, the following results and discussion sections
regarding physical effects on the $^{234}$Th flux estimates is based on the steady state model only.

3.4 Export fluxes of $^{234}$Th
Fluxes of $^{234}$Th due to radioactive production and decay (hereafter 'production flux'),
upwelling, and vertical diffusion were reported in Table 1 and Figure 6 for both depths 5-20
m below the ML and at 100 m. The production fluxes of $^{234}$Th at 5-20 m below the ML

ranged from 560 dpm m$^{-2}$ d$^{-1}$ to 1880 dpm m$^{-2}$ d$^{-1}$, whereas at 100 m they were much higher at 850 dpm m$^{-2}$ d$^{-1}$ to 3370 dpm m$^{-2}$ d$^{-1}$. There is no discernable trend regarding the production fluxes between the shelf and offshore stations, similar to those seen along the eastern GP16 transect (Black et al. 2017).

Alongshore winds were unusually weak off Peru preceding and during our sampling campaign as a result of the 2017 coastal El Niño (Echevin et al., 2018; Lüdke et al., in review 2020; Peng et al., 2019), which resulted in nominal upwelling in the water column. At nearshore stations, upwelling rates at the base of the ML varied between $1.3 \times 10^{-7}$ m s$^{-1}$ and $9.7 \times 10^{-6}$ m s$^{-1}$, whereas upwelling rates at offshore stations were on the order of $10^{-10}$ m s$^{-1}$ to $10^{-8}$ m s$^{-1}$ and essentially negligible. As a result, upwelled $^{234}$Th fluxes at 5-20 m below the ML were only significant at stations closest to shore; these stations were 428 (130 dpm m$^{-2}$ d$^{-1}$), 883-12 (80 dpm m$^{-2}$ d$^{-1}$) and 904-16 (280 dpm m$^{-2}$ d$^{-1}$) whose upwelled $^{234}$Th fluxes accounted for 10%, 11% and 25% of the total $^{234}$Th fluxes respectively (Figure 6). Upwelled $^{234}$Th fluxes at the rest of the stations accounted for less than 2% of the total $^{234}$Th fluxes (6% at stations 353 and 907-11) and were insignificant. At 100 m, both vertical $^{234}$Th gradients and upwelling rates were significantly smaller compared to shallower depths. As a result, upwelled $^{234}$Th fluxes were less than 70 dpm m$^{-2}$ d$^{-1}$, or less than 4% of total $^{234}$Th fluxes.

Similarly, vertical diffusivities, shown as running mean over 20 m in Figure S1, were an order of magnitude higher at shallow stations ($3.2 \times 10^{-4} \pm 1.7 \times 10^{-4}$ m$^2$ s$^{-1}$; 1SD, 27 m to 100 m below sea surface) compared to those at deep stations ($1.7 \times 10^{-5} \pm 0.6 \times 10^{-5}$ m$^2$ s$^{-1}$; 1SD; 34 – 100 m below sea surface). Within the upper 27 m to 33 m layer at offshore deep stations, vertical diffusivities decreased exponentially by an order of magnitude within a few meters; below this depth, vertical diffusivities remained relatively stable (Figure S1). This is not surprising as wind-driven turbulent is most significant at the ocean surface (Buckingham et al., 2019). In this study, the sampling depths immediately below the ML were generally 30

m and 60 m. A few high vertical diffusivity values around 30 m at deep stations were unlikely

representative for the 30 m – 60 m water column layer. We thus opted to only apply vertical

diffusivities below 33 m at deep stations. Relative standard errors (RSE) associated with

diffusivity estimates varied from 35% to 55%. Vertical diffusive $^{234}$Th fluxes at 5-20 m below

the ML, determined using both vertical diffusivity and vertical $^{234}$Th gradient, varied greatly

amongst stations. At shallow stations 428, 458, and 883-12, vertical diffusive $^{234}$Th fluxes

made up 37% (490 dpm m$^{-2}$ d$^{-1}$), 14% (160 dpm m$^{-2}$ d$^{-1}$), and 21% (160 dpm m$^{-2}$ d$^{-1}$) of total

$^{234}$Th fluxes, respectively (Figure 6). At the rest of the stations, vertical diffusive $^{234}$Th fluxes

appeared to be insignificant, ranging between 1% and 10% in the total $^{234}$Th flux budget. At

100 m, vertical diffusive $^{234}$Th fluxes at station 428, 458, and 883-12 remained high at 390

dpm m$^{-2}$ d$^{-1}$, 150 dpm m$^{-2}$ d$^{-1}$, 120 dpm m$^{-2}$ d$^{-1}$, respectively, whereas those at the rest of the

stations accounted for < 2% of the total $^{234}$Th flux.

Horizontal advective and diffusive $^{234}$Th fluxes were both very small. Average

alongshore current velocities (Lüdke et al., in review 2020) for the top layer varied from 0.06

m s$^{-1}$ to 0.34 m s$^{-1}$. At the peripheral of a freshly-formed anticyclonic eddy (St. 915-1),

alongshore current velocities could be as high as 0.53 m s$^{-1}$. Taking the mean alongshore

velocity of 0.2 m s$^{-1}$ and the net difference in alongshore $^{234}$Th gradient of 2 x 10$^{-7}$ dpm L$^{-1}$ m$^{-1}$, the resulting net horizontal advective $^{234}$Th flux at the top layer is ~ 50 dpm m$^{-2}$ d$^{-1}$, a mere

3-9% of the total $^{234}$Th fluxes.

Horizontal diffusive $^{234}$Th flux was estimated using an average eddy diffusivity of

1000 m$^2$ s$^{-1}$ (see Methods section 2.3.3) and the alongshore $^{234}$Th gradient. A maximum value

of 10 dpm m$^{-2}$ d$^{-1}$ was calculated, which accounted for <1% of total $^{234}$Th flux at all stations.

Note that the horizontal advective and lateral diffusive fluxes presented here are a rough

estimate and should only provide an idea of their order of magnitude. Due to the uncertainty

inherent to the estimates, we refrain from adding these values to Table 1.

## 4. Discussion

4.1 Lack of linear $^{238}$U – salinity correlation in the Peruvian OMZ

The water column profiles of $^{238}$U in the Peruvian OMZ (Figure 2) are similar to those seen in the open ocean (see compilations in Owens et al., 2011 and Van Der Loeff et al. (2006), and references therein). It thus appears that water column suboxic/anoxic conditions alone is not sufficient to remove U, in contrast to sedimentary U studies underlying low oxygen waters where soluble U(VI) diffused downward into subsurface sediments and reduced to insoluble U(IV) (Anderson et al., 1989; Böning et al., 2004; Scholz et al., 2011). Our inference is in accord with water column $^{238}$U studies in intense OMZs in the eastern tropical North Pacific (Nameroff et al., 2002) and the Arabian Sea (Rengarajan et al., 2003), where $^{238}$U concentrations remain constant over the entire upper water column studied.

Dissolved $^{238}$U and salinity across the entire Peruvian OMZ displayed poor linear correlation regardless of seawater oxygen concentrations (Figure 7a-b). The general consensus is that U behaves conservatively in oxic seawater in the open ocean and early observations have shown that $^{238}$U activities can be calculated from salinity based on a simple linear correlation between the two (e.g. Chen et al., 1986; Ku et al., 1977). Compilations in Van Der Loeff et al. (2006) and Owens et al. (2011) further demonstrated that the majority of uranium data points in the global seawater dataset follow a linear correlation with seawater salinity. The $^{238}$U-salinity formulations from either Chen et al. (1986) or Owens et al. (2011) are thus generally appropriate for open ocean conditions and have been widely used in $^{234}$Th flux studies. However, this linear $^{238}$U-salinity correlation breaks down in the Peruvian OMZ. Furthermore, the measured $^{238}$U activities in this study correlated poorly with those calculated from salinity using the Owens formulation regardless of water column oxygen concentrations (Table S2, Figure 7c), with the former significantly higher than the projected values and

differences up to 10%. Both evidences suggested that non-conservative processes have
introduced significant amount of dissolved U into the water column.

It is likely that this poor $^{238}$U-salinity correlation in the water column is not a unique

feature off the coast of Peru. Poor correlations between dissolved $^{238}$U and salinity have been
previously observed in open ocean settings such as the Arabian Sea (Rengarajan et al., 2003)
and the Pacific Ocean (Ku et al., 1977), and shelf-estuary systems such as the Amazon shelf
(McKee et al., 1987; Swarzenski et al., 2004). It is possible that the narrow range of salinity
within any single ocean basin precludes a meaningful $^{238}$U-salinity correlation (Ku et al.,
1977; Owens et al., 2011). For the Peruvian shelf system, two possible scenarios may further
explain the lack of linear $^{238}$U-salinity correlation in the water column. Firstly, authigenic U
within the sediments may be remobilized under ENSO-related oxygenation events. In
reducing pore water, U reduction and removal from pore water is usually seen within the Fe
reduction zone (Barnes and Cochran, 1990; Barnes and Cochran, 1991; Scholz et al., 2011).
As such, a downward diffusive flux of U across the water-sediment interface is expected in
reducing sedimentary environment. However, pore water and bottom water geochemistry
measurements during two previous cruises (M77-1 and M77-2) along an 11°S transect off
Peru showed large diffusive fluxes of U out of the Peruvian shelf sediments despite that both
Fe reduction and U reduction took place in the top centimeters of sediments (Scholz et al.,
2011). It was suggested that a minute increase in bottom water oxygen concentration induced
by El Niño events would be sufficient in shifting the U(VI)/U(IV) boundary by a few
centimeters and remobilize authigenic U (Scholz et al., 2011). Preceding and during our
sampling campaign, a coastal El Niño event, with coastal precipitation as strong as the 1997-
98 El Niño event, had developed rapidly and unexpectedly in January, and disappeared by
May 2017 during cruise M136 (Echevin et al., 2018; Garreaud, 2018; Peng et al., 2019). This
strong coastal El Niño event could induce an oxygenation event large enough to remobilized
authigenic U along the Peruvian shelf. Secondly, resuspension of bottom sediments and
subsequent desorption of U from ferric-oxyhydroxides could affect the $^{238}$U-salinity
relationship, similar to that seen on the Amazon shelf at salinity above 10 (McKee et al.,
1987) and in laboratory experiments (Barnes and Cochran, 1993). Fe reduction and release
from the Peruvian shelf sediments (Noffke et al., 2012; Scholz et al., 2014) could release
additional U to overlying waters. The magnitude of such, however, has not been quantified.

The consequence of the notable difference between measured $^{238}$U in this study and

salinity-based $^{238}$U to $^{234}$Th flux according to Eq. (2) is neither linear nor straightforward,
because the vertical gradients of both $^{238}$U and $^{234}$Th strongly affects the impacts of $^{238}$U
variations on $^{234}$Th fluxes. In this study, $^{234}$Th fluxes at 100 m derived from salinity-based
$^{238}$U lead to significant underestimation of $^{234}$Th fluxes by an average of 20% and as high as
40% (Table 2). These differences in $^{234}$Th fluxes will have direct consequences for $^{234}$Th
derived elemental fluxes such as C, N, P and trace metals. It is thus important to note that U
concentrations in coastal systems are highly sensitive to bottom water oxygen concentrations
and redox-related U addition, variability of which is expected to intensify with future climate
change (Shepherd et al., 2017). Relatively minor variations in dissolved $^{238}$U could account
for substantial overestimation/underestimation of the depth-integrated $^{234}$Th fluxes. We thus
encourage future $^{234}$Th flux studies in such environments to include seawater $^{238}$U analysis.

4.2 Dynamic advective and diffusive $^{234}$Th fluxes

The significance of advection and diffusion in the total $^{234}$Th flux budget highly

depends on the upwelling rate, current velocity, vertical diffusivity, and $^{234}$Th gradient on the
horizontal and vertical directions. Our results demonstrated that physical processes off Peru
during and post the 2017 coastal El Niño have very limited impact on the downward fluxes of
$^{234}$Th (Figure 6).
Our findings are in reasonable agreement with those from the GEOTRACES GP16
eastern section along 12°S from Peru to Tahiti, in which Black et al. (2018) quantified both
horizontal and vertical advective $^{234}$Th fluxes. Horizontal advective fluxes for the upper 30 m
water column estimated during GP16 were ~180 dpm m$^{-2}$ d$^{-1}$ for all nearshore and offshore
stations, similar in magnitude to those estimated in our study (~50 dpm m$^{-2}$ d$^{-1}$). Upwelling
fluxes along GP16 eastern section was suggested to account for 50% to 80% of total $^{234}$Th
fluxes at the base of the euphotic zone (Black et al., 2018), a depth similar to or slightly
deeper than ML depths in the current study where upwelling fluxes accounted for less than
25% of total $^{234}$Th fluxes). Total $^{234}$Th fluxes along the GP16 eastern section, ranging from
4000 to 5000 dpm m$^{-2}$ d$^{-1}$ at the base of the euphotic zone, were much higher than those in our
study (560 to 1900 dpm m$^{-2}$ d$^{-1}$ 5-20 m below the ML). This difference could be related to the
period of sampling (austral autumn and winter 2017 in our study *vs.* austral spring 2013 for
the GP16 section). We note that the estimated vertical mixing rates based on $^7$Be isotope at
the base of the euphotic zone along the GP16 section (Kadko, 2017) were at least an order of
magnitude higher than the upwelling rates at the base of the ML at nearby stations in our
study. This difference could stem from different methods used to estimate upwelling rates at
different timescales, and may also reflect the dynamic upwelling system off Peru in which
upwelling rates vary greatly seasonally and interannually. During cruises M136 and M138,
upwelling favorable easterly winds off Peru were weak, resulting in negligible coastal
upwelling. Coastal upwelling in the same general area was also suggested to be negligible in
austral summer 2013 during cruise M92 due to nominal surface wind stress (Thomsen et al.,
2016). Results from studies conducted in the same year (October to December 2013, Kadko,
2017;  December 2012, Steinfeldt et al., 2015; January 2013, Thomsen et al., 2016) indicate
that seasonal upwelling rates vary drastically in the Peruvian upwelling zone. The seasonal
dynamics of coastal upwelling off Peru are similar to those seen in the Arabian Sea, where
large upwelled $^{234}$Th fluxes only occurred during mid-late southwest monsoon at stations
close to shore (Buesseler et al., 1998). Our findings lend further support to earlier studies that
advection and diffusion are seasonally important for $^{234}$Th fluxes in regions with high
upwelling velocities and diffusivities such as the equatorial Pacific (Bacon et al., 1996;
Buesseler et al., 1995; Dunne and Murray, 1999) and coastal sites such as the Arabian Sea
(Buesseler et al., 1998) and offshore Peru (Black et al., 2018; this study).

4.3 Residence time of $^{234}$Th in the Peruvian OMZ
The residence time calculated using equation (6) was based on a simplified one-
dimension (1D) model of Zimmerman (1976). This 1D steady state model is obviously an
oversimplification of a multi-dimensional process, it however provides a good first order
estimate for understanding the highly dynamic nature of the $^{234}$Th residence time. It also
provides a reasonable value that can be directly compared to values estimated in earlier $^{234}$Th
flux studies that did not consider the physical processes. Furthermore, we showed in the
Discussion (sections 4.2) that physical processes, namely upwelling and vertical diffusion, are
only important at a few shelf stations. We thus consider this simple 1D model robust in
estimating the residence time of total $^{234}$Th.
In this study, residence time of total $^{234}$Th in the top layer varied from 20 days at
shallow stations to 95 days at deep stations (mean $\tau = 51 \pm 23$ days, 1SD, n = 24; Table 3).
These values were similar to those estimated within the California Current (Coale and
Bruland, 1985) and the residence times of particulate organic carbon (POC) and nitrogen
(PON) (Murray et al., 1989), but were much longer than predicted in nearshore shelf waters
where residence times of total $^{234}$Th were on the order of a few days (Kaufman et al., 1981;
Kim et al., 1999; and references therein). The longer residence times estimated in our study
could reflect a combination of weak surface $^{234}$Th deficits ($^{234}$Th = 0.63 to 1.82 dpm L$^{-1}$)
(Figure 3) and low export fluxes (800 to 2000 dpm m$^{-2}$ d$^{-1}$, Figure 7). Nearshore seawater
samples during GP16 (Black et al., 2018) featured similar surface $^{234}$Th deficits ($^{234}$Th = 0.63
to 1.33 dpm L$^{-1}$) but much higher downward $^{234}$Th fluxes (4000 to 5000 dpm m$^{-2}$ d$^{-1}$) as a
result of strong upwelling, implying that residence times of total $^{234}$Th in the Peruvian OMZ
during GP16 occupation would be 3 – 6 times shorter. Indeed, a quick re-assessment of the
GP16 data predicted a shorter residence time of total $^{234}$Th of 5 – 23 days within the euphotic
zone of the coastal Peruvian OMZ.

These temporal variations on the residence times of total $^{234}$Th have important

implications for the estimation of POC fluxes and quantification of carbon export efficiency.
Firstly, seasonal changes in Th residence times reflect variations in particle removal over
different integrated timescales. For example, POC produced in surface waters during GP16
(austral spring 2013) (Black et al., 2018) would have been exported out of the euphotic zone
3-6 times faster than it did during austral autumn 2017 (this study). Secondly, to properly
evaluate carbon export efficiency, surface net primary production (NPP) should be averaged
over a similar timescale as the residence time of total $^{234}$Th during station occupation.
Applying a 16-day averaged NPP for export efficiency estimate (Black et al., 2018; Henson et
al., 2011) would likely not be appropriate in the current study in which total $^{234}$Th fluxes
integrated timescales of several weeks. $^{234}$Th residence times should thus be properly
quantified in coastal studies before deriving export efficiencies over varying NPP integration
timescales.

**5. Conclusions and implications for coastal $^{234}$Th flux studies**

Advection and diffusion are important in coastal and upwelling regions with respect to

$^{234}$Th export fluxes (Bacon et al., 1996; Buesseler et al., 1995; Dunne and Murray, 1999;
Buesseler et al., 1998). Our findings show that their significance is subject to the seasonal
variability of the current and upwelling velocities, diffusivities and $^{234}$Th gradients, and
should be evaluated on a case-to-case basis. Advective fluxes are perhaps the most
straightforward to estimate as current velocities can be obtained routinely from shipboard
ADCP measurements and upwelling rates calculated from satellite wind stress (Steinfeldt et
al., 2015; Bacon et al., 1996). Horizontal and vertical velocities derived from general ocean
circulation models also provide a good first order estimate for advective [234]Th fluxes; this
approach has been successfully demonstrated in a few studies (Buesseler et al., 1995;
Buesseler et al., 1998). In addition, the anthropogenic SF$_6$ tracer and radium isotopes, widely
used to quantify nutrient and Fe fluxes (Charette et al., 2007; Law et al., 2001), as well as [7]Be
isotope (Kadko, 2017), could be used independently to constrain horizontal and vertical
exchange rates of [234]Th (Morris et al., 2007; Charette et al., 2007; Buesseler et al., 2005).
When *in situ* microstructure measurements are available (this study), vertical diffusivity can
be directly calculated to estimate the vertical diffusive [234]Th fluxes. Yet, microstructure
analysis is not a routine measurement on oceanographic cruises. Earlier studies in the
equatorial Pacific and the Gulf of Maine have shown that general ocean circulation models
and a simple assumption on dissipation coefficients could provide a robust estimate on
vertical and horizontal diffusivities (Benitez-Nelson et al., 2000; Gustafsson et al., 1998;
Charette et al., 2001). Therefore, the calculation of physical fluxes is possible, though
challenging, and [234]Th fluxes due to physical processes should be carefully considered when
conducting research in a coastal and upwelling systems.
A striking finding in this study is that the assumption of a linear [238]U-salinity
correlation could lead to one of the largest errors in [234]Th flux estimates. In our study, using
the salinity-based [238]U activities resulted in significant underestimation of total [234]Th fluxes by
as much as 40%. Because the translation of [238]U activities to [234]Th fluxes is not linear, larger
differences between measured and salinity-based [238]U do not necessarily contribute to greater
overestimation or underestimation of [234]Th fluxes. For example, moderate difference of 3-6%
in $^{238}$U throughout the upper 100 m at station 898 lead to 40% difference in final $^{234}$Th flux,
while a 5-9% difference in $^{238}$U at station 906 only resulted in 16% $^{234}$Th flux difference
(Table 2, S2). We would thus stress the importance of $^{238}$U measurements in future $^{234}$Th flux
studies particularly in coastal and shelf regions.
Finally, our study showed that the residence times of total $^{234}$Th in the Peruvian
nearshore waters varied seasonally. Tropical OMZs are important hotspots for carbon
sequestration from the atmosphere and enhanced sedimentary carbon preservation (Arthur et
al., 1998; Suess et al., 1987). These OMZs are projected to intensify as a result of future
climate change (Keeling and Garcia, 2002; Schmidtko et al., 2017; Stramma et al., 2008).
Future studies should take into consideration the large temporal variations of the residence
times of total $^{234}$Th in order to properly evaluates how carbon biogeochemical cycles and
carbon export efficiency in these OMZs will respond to continuing ocean deoxygenation,

**Data availability**
Data are available in supplementary tables and archived  at
https://doi.org/10.1594/PANGAEA.921917 (Xie et al., 2020).

**Author contribution**
RCX, FACLM and EAP designed the study. RCX carried out sampling, on-board beta
counting of $^{234}$Th, and drafted the manuscript. IR conducted $^{234}$Th and $^{238}$U analyses at home
laboratory. JL computed current velocities and vertical diffusivities respectively from
VmADCP and microstructure profiler data. All co-authors had a chance to review the
manuscript and contributed to discussion and interpretation of the data presented.

**Competing interests**


The authors declare that they have no conflict of interest.

**Acknowledgements**


We thank the crew and science party on board M136 and M138 for their help in sample
collection and instrument operation. Thank you to SiaoJean Ko, Dominik Jasinski, André
Mutzberg and Mario Esposito for their laboratory assistance. We thank two anonymous
reviewers and the associate editor, Marilaure Grégoire, for their constructive comments. The
project, cruises, IR, JL and RCX were funded by the German SFB 754 program ('Climate-
Biogeochemistry Interactions in the Tropical Ocean'), RCX additionally by a DFG research
grant (project number 432469432), and FACLM by a DFG Fellowship of the Excellence
Cluster "The Future Ocean" (CP1403). This manuscript benefited from stimulating
discussions at the BIARRITZ ('bridging international activity and related research into the
twilight zone') workshop held in Southampton, UK in 2019.

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

Figure captions
Figure 1. Maps showing (a) locations of each station from M136 (white squares) and M138
(grey circles) and (B) monthly-averaged current field in the top 15 m from April 16 to May
15, 2017 derived from altimetry measurements (http://marine.copernicus.eu/; product ID:
MULTIOBS_GLO-PHY_REP_015_004). Color boxes in (a) schematically divide the four
shelf-offshore transects. Map (a) was created with Ocean Data View (Schlitzer, 2014). The
white box in (b) highlights our study area.

Figure 2. Profiles of $^{238}$U (black) and $^{234}$Th (orange squares – M136; orange circles – M138)
along with concentrations of oxygen (grey) and fluorescence (green). Profiles are organized
by cruises, transects, and distance to shore from left to right and top to bottom, indicated by
east (E) to west (W) arrows. Error bars for both $^{238}$U and $^{234}$Th are indicated. Red dashed lines
indicate the depth of the mixed layer. The start of the oxygen deficient zone is where oxygen
diminishes. Bottom depths are indicated for stations whose bottom depths are shallower than
600 m.

Figure 3. Shelf-offshore distributions of $^{234}$Th/$^{238}$U along the four studied transects, as shown
in Figure 1, for M136 (left) and M138 (right). White dots denote station location.

Figure 4. Distributions of averaged $^{234}$Th activities during M136 (a, top 30 m) and M138 (b,
top 50 m).

Figure 5. Profiles of temperature (solid lines) and salinity (dashed lines) for (a) repeated
stations 458 (purple) and 508 (yellow), and (d) 495 (blue) and 516 (orange); (b) and (c)
respectively profiles for stations 458 and 508 of $^{238}$U (black), $^{234}$Th (color squares), and
concentrations of oxygen (grey) and fluorescence (green). (e) and (f) respectively profiles for
stations 495and 516 of $^{238}$U (black), $^{234}$Th (color squares), and concentrations of oxygen
(grey) and fluorescence (green).

Figure 6. Bar charts of $^{234}$Th fluxes due to production and decay (blue), upwelling (orange),
and vertical diffusion (grey) for the depths at 5 – 20 m below the ML (top) and 100 m below
sea surface (bottom). Color boxes corresponds to individual transects in Figure 1. Within each
transect stations from west (offshore) to east (nearshore) are listed from left to right. Error
bars (1SE) are indicated.

Figure 7. Cross plots of measured $^{238}$U activities vs. salinity for M136 (a) and M138 (b),
showing poor linear relationship between $^{238}$U and salinity. (c) shows a direct comparison
between measured and salinity-based $^{238}$U to further highlight the large difference between the
two. The solid blue line indicates the 1:1 ratio between measured and projected $^{238}$U. Blue
dashed lines indicate the ± errors reported in Owens et al. (2011). Error bars for measured
$^{238}$U activities are smaller than symbols.

**Table 1.** $^{234}$Th fluxes due to production and decay, upwelling and vertical diffusion below the mixed layer and at 100 m. Horizontal advective fluxes were not quantified at 100 m. Refer to text for details.

| Cruise | Station | Cast | Mixed layer depth (m) | Upper oxycline depth (m) | Maximum fluorescence (µg L⁻¹) | Equilibrium depth (m) | $^{234}$Th flux at the base of the ML | | | | | | $^{234}$Th flux at 100 m | | | | |
|---|---|---|---|---|---|---|---|---|---|---|---|---|---|---|---|---|---|
| | | | | | | | Depth (m) | Production and decay (dpm m⁻² d⁻¹) | Upwelling (dpm m⁻² d⁻¹) | Diffusion (dpm m⁻² d⁻¹) | Final flux (dpm m⁻² d⁻¹) | 1 SD (dpm m⁻² d⁻¹) | Production and decay (dpm m⁻² d⁻¹) | Upwelling (dpm m⁻² d⁻¹) | Diffusion (dpm m⁻² d⁻¹) | Final flux (dpm m⁻² d⁻¹) | 1 SD (dpm m⁻² d⁻¹) |
| M136 | 353 | 1 | 25 | 102 | 1.20 | 100 | 30 | 907 | 52 | -36 | 923 | 69 | 1422 | -14 | 2 | 1410 | 189 |
| M136 | 380 | 1 | 26 | 129 | 0.87 | 80 | 30 | 1145 | 0 | -41 | 1105 | 54 | 1637 | 0 | -1 | 1637 | 132 |
| M136 | 402 | 1 | 24 | 129 | 7.51 | 100 | 30 | 808 | 0 | -75 | 732 | 64 | 1234 | 0 | 2 | 1236 | 111 |
| M136 | 428 | 1 | 10 | 76 | 4.11 | 30 | 30 | 983 | -128 | 493 | 1348 | 129 | 1772 | 33 | -390 | 1415 | 256 |
| M136 | 445 | 1 | 17 | 64 | 2.07 | 100 | 30 | 820 | -10 | 16 | 826 | 66 | 1621 | 53 | 6 | 1681 | 165 |
| M136 | 458 | 1 | 5 | 55 | 1.61 | 100 | 30 | 1012 | -18 | 161 | 1155 | 117 | 2101 | -11 | 145 | 2235 | 238 |
| M136 | 472 | 1 | 11 | 29 | 7.41 | 200 | 40 | 1887 | 15 | -29 | 1872 | 77 | 3315 | -12 | 63 | 3366 | 233 |
| M136 | 495 | 1 | 18 | 50 | 1.13 | 200 | 30 | 1149 | 1 | -19 | 1130 | 50 | 3195 | 2 | -5 | 3192 | 89 |
| M136 | 516 | 1 | 16 | 45 | 3.77 | 200 | 30 | 614 | 0 | 1 | 615 | 49 | 2229 | 2 | -4 | 2227 | 109 |
| M136 | 547 | 1 | 22 | 48 | 1.28 | 150 | 30 | 791 | 0 | 85 | 877 | 61 | 2510 | 0 | -15 | 2495 | 118 |
| M136 | 559 | 1 | 20 | 79 | 1.70 | 85 | 50 | 623 | 3 | -67 | 559 | 117 | 854 | -4 | 2 | 852 | 120 |
| M136 | 567 | 1 | 21 | 50 | 2.40 | 150 | 30 | 1593 | 0 | -23 | 1570 | 52 | 3011 | 0 | -11 | 3000 | 86 |
| M138 | 879 | 3 | 43 | 93 | 2.24 | 200 | 60 | 1249 | 0 | -16 | 1266 | 91 | 1702 | 0 | -5 | 1697 | 111 |
| M138 | 882 | 10 | 39 | 211 | 2.68 | 150 | 50 | 1321 | -7 | 16 | 1331 | 63 | 2264 | 19 | -12 | 2272 | 82 |
| M138 | 883 | 12 | 10 | 220 | 1.31 | 250 | 30 | 683 | -84 | -159 | 758 | 108 | 1782 | 31 | -121 | 1692 | 179 |
| M138 | 888 | 7 | 41 | 127 | 1.59 | 150 | 50 | 1364 | 0 | -120 | 1244 | 62 | 1813 | 0 | -4 | 1809 | 86 |
| M138 | 892 | 14 | 47 | 128 | 1.05 | 100 | 60 | 1395 | 33 | -118 | 1309 | 72 | 1743 | -3 | 1 | 1741 | 99 |
| M138 | 898 | 1 | 38 | 101 | 1.42 | 60 | 50 | 1099 | 0 | -19 | 1080 | 104 | 1091 | 0 | 0 | 1091 | 125 |
| M138 | 904 | 16 | 12 | 72 | 3.63 | 150 | 20 | 812 | 275 | 0 | 1087 | 76 | 2643 | 0 | -9 | 2634 | 79 |
| M138 | 906 | 18 | 32 | 81 | 1.73 | 200 | 40 | 1796 | 0 | 4 | 1799 | 41 | 3100 | 0 | -1 | 3100 | 77 |
| M138 | 907 | 11 | 31 | 100 | 1.29 | 60 | 60 | 1594 | -88 | 13 | 1518 | 147 | 1787 | 67 | -2 | 1853 | 140 |
| M138 | 912 | 3 | 37 | 70 | 2.75 | >600 | 50 | 1960 | 0 | -79 | 1881 | 43 | 2975 | 0 | -3 | 2972 | 78 |
| M138 | 915 | 1 | 26 | 99 | 3.51 | 200 | 40 | 1628 | 0 | 22 | 1650 | 38 | 2752 | 0 | 0 | 2752 | 93 |
| M138 | 919 | 1 | 19 | 79 | 4.46 | 150 | 30 | 1316 | 0 | 49 | 1365 | 32 | 3249 | 0 | -8 | 3241 | 85 |

**Table 2. Comparison of <sup>234</sup>Th fluxes at 100 m calculated with measured**
**<sup>238</sup>U activities and those with salinity-based <sup>238</sup>U.**

**Table 2. Comparison of $^{234}$Th fluxes at 100 m calculated with measured $^{238}$U activities and those with salinity-based $^{238}$U.**

| Cruise | Station | Cast | $^{234}$Th fluxes at 100 m* | | Difference |
| | | | measured | predicted | |
| | | | dpm m$^{-2}$ d$^{-1}$ | dpm m$^{-2}$ d$^{-1}$ | % |
| --- | --- | --- | --- | --- | --- |
| M136 | 353 | 1 | 1422 | 1320 | 8 |
| M136 | 380 | 1 | 1637 | 1304 | 26 |
| M136 | 402 | 1 | 1234 | 865 | 43 |
| M136 | 428 | 1 | 1772 | 1443 | 23 |
| M136 | 445 | 1 | 1621 | 1365 | 19 |
| M136 | 458 | 1 | 2101 | 1859 | 13 |
| M136 | 472 | 1 | 3315 | 3073 | 8 |
| M136 | 495 | 1 | 3195 | 3058 | 4 |
| M136 | 516 | 1 | 2229 | 2140 | 4 |
| M136 | 547 | 1 | 2510 | 2313 | 9 |
| M136 | 559 | 1 | 854 | 751 | 14 |
| M136 | 567 | 1 | 3011 | 2879 | 5 |
| | | | | | |
| M138 | 879 | 3 | 1702 | 1515 | 12 |
| M138 | 882 | 10 | 2264 | 1875 | 21 |
| M138 | 883 | 12 | 1782 | 1352 | 32 |
| M138 | 888 | 7 | 1813 | 1441 | 26 |
| M138 | 892 | 14 | 1743 | 1257 | 39 |
| M138 | 898 | 1 | 1091 | 770 | 42 |
| M138 | 904 | 16 | 2643 | 2280 | 16 |
| M138 | 906 | 18 | 3100 | 2673 | 16 |
| M138 | 907 | 11 | 1787 | 1308 | 37 |
| M138 | 912 | 3 | 2975 | 2572 | 16 |
| M138 | 915 | 1 | 2752 | 2380 | 16 |
| M138 | 919 | 1 | 3249 | 2862 | 14 |

* For comparison purposes, we only report here $^{234}$Th fluxes due to radioactive production and decay.

**Table 3. Residence time of total** [234]**Th in the top layers of Peruvian OMZ.**

| Cruise | Station | Cast | Average [234]Th in the top layer* dpm L[-1] | Residence time days |
|--------|---------|------|--------------------------------|---------------------|
| M136 | 353 | 1 | 1.48 | 46 |
| M136 | 380 | 1 | 1.35 | 35 |
| M136 | 402 | 1 | 1.64 | 61 |
| M136 | 428 | 1 | 1.57 | 35 |
| M136 | 445 | 1 | 1.64 | 61 |
| M136 | 458 | 1 | 1.45 | 38 |
| M136 | 472 | 1 | 0.93 | 20 |
| M136 | 495 | 1 | 1.20 | 31 |
| M136 | 516 | 1 | 1.74 | 85 |
| M136 | 547 | 1 | 1.67 | 63 |
| M136 | 559 | 1 | 1.75 | 94 |
| M136 | 567 | 1 | 1.41 | 45 |
| | | | | |
| M138 | 879 | 3 | 1.59 | 75 |
| M138 | 882 | 10 | 1.81 | 69 |
| M138 | 883 | 12 | 1.87 | 74 |
| M138 | 888 | 7 | 1.68 | 67 |
| M138 | 892 | 14 | 1.69 | 65 |
| M138 | 898 | 1 | 1.66 | 92 |
| M138 | 904 | 16 | 1.32 | 24 |
| M138 | 906 | 18 | 1.15 | 25 |
| M138 | 907 | 11 | 1.04 | 41 |
| M138 | 912 | 3 | 1.25 | 33 |
| M138 | 915 | 1 | 1.16 | 28 |
| M138 | 919 | 1 | 1.17 | 26 |

* Here 'the top layer' refers to the top 30 m during M136 and top 50 m during M138.

Figure 1

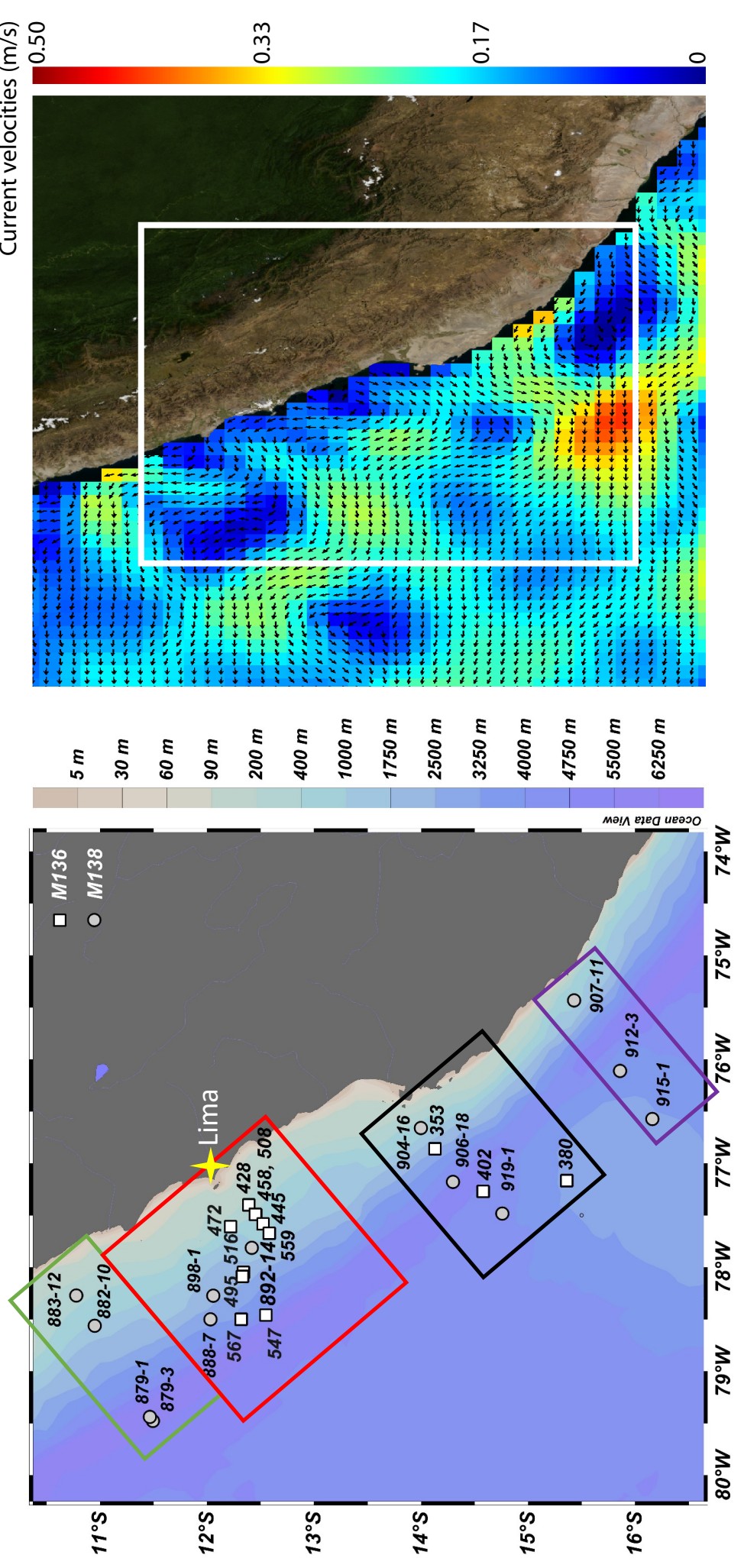

Figure 2

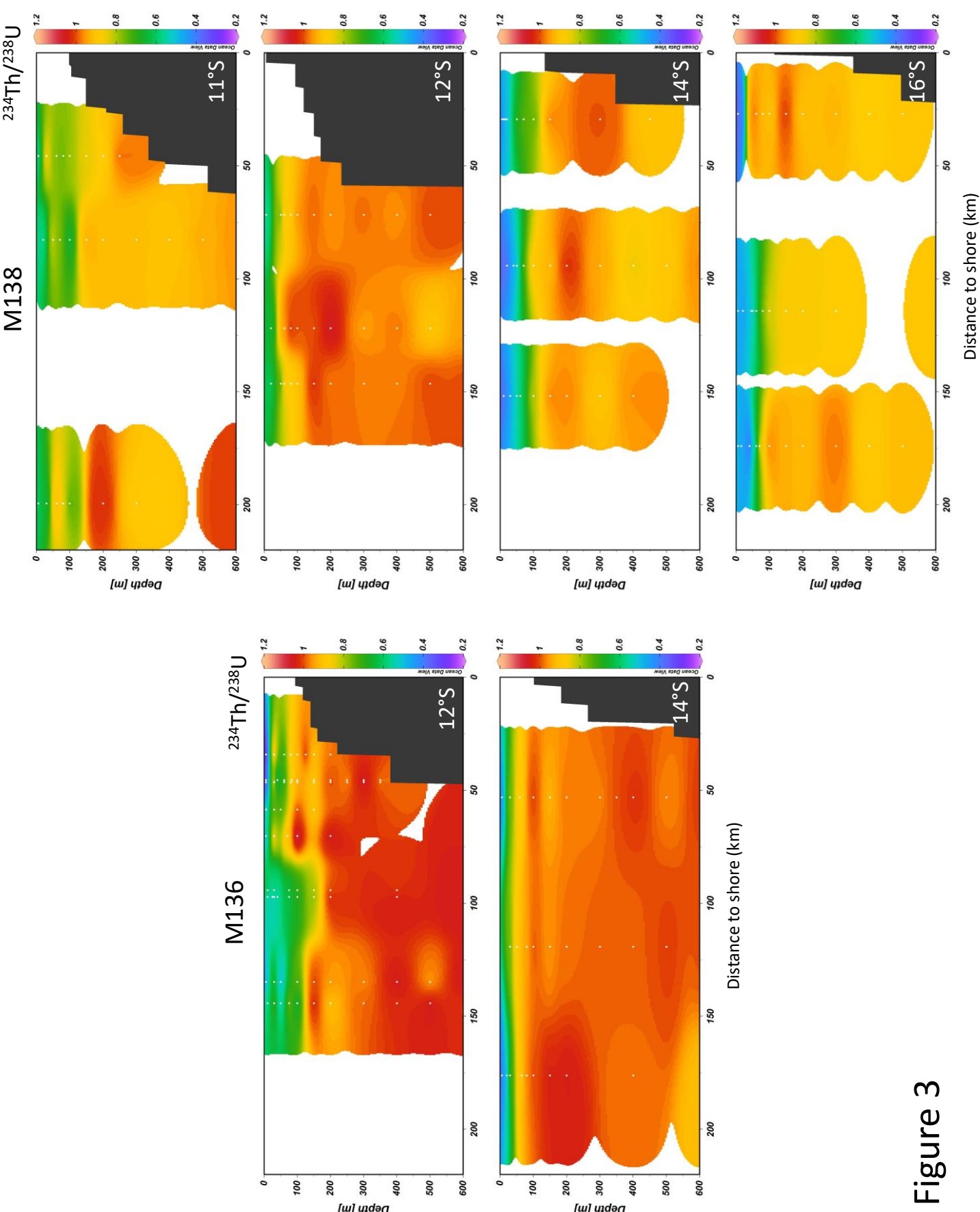

Figure 3

Figure 4

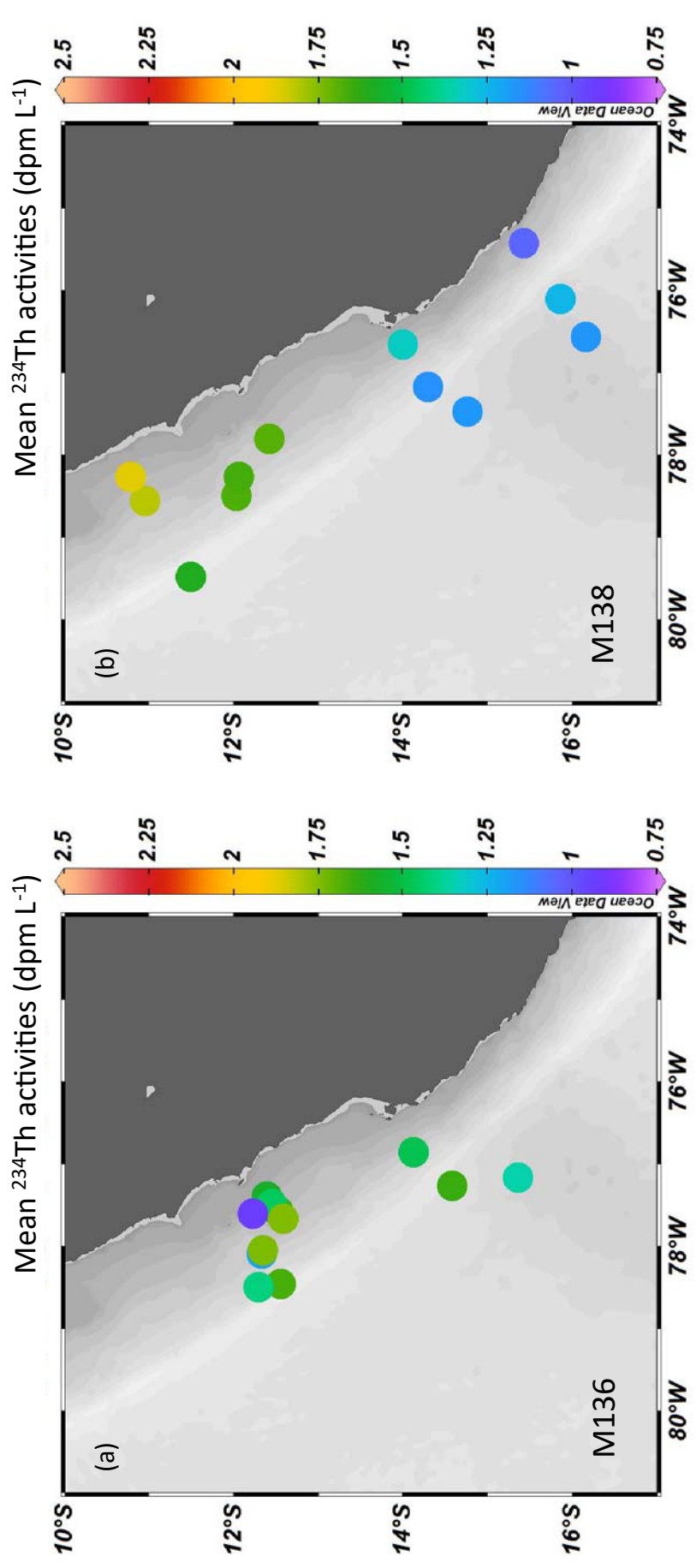

Figure 5

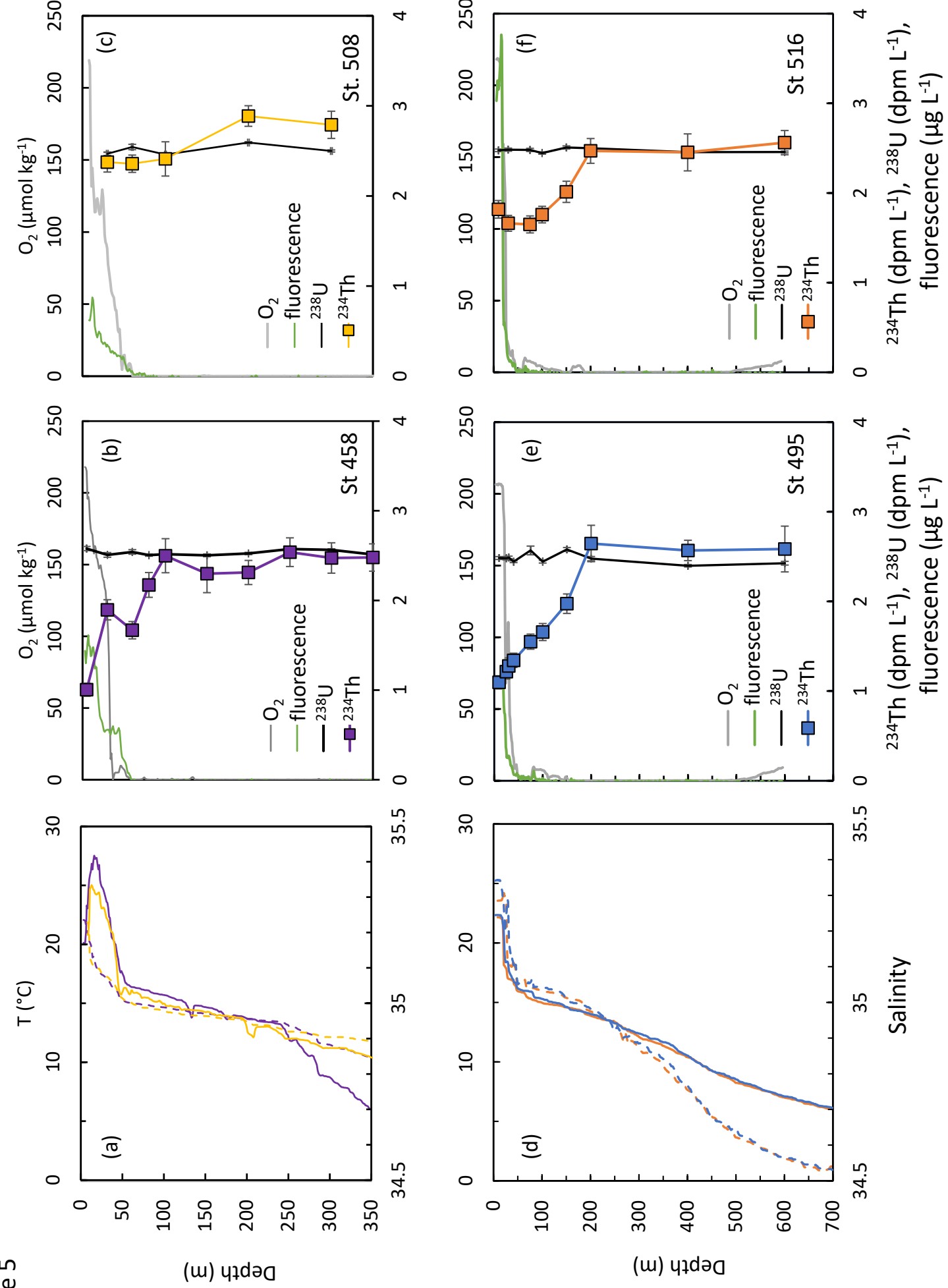

Figure 6

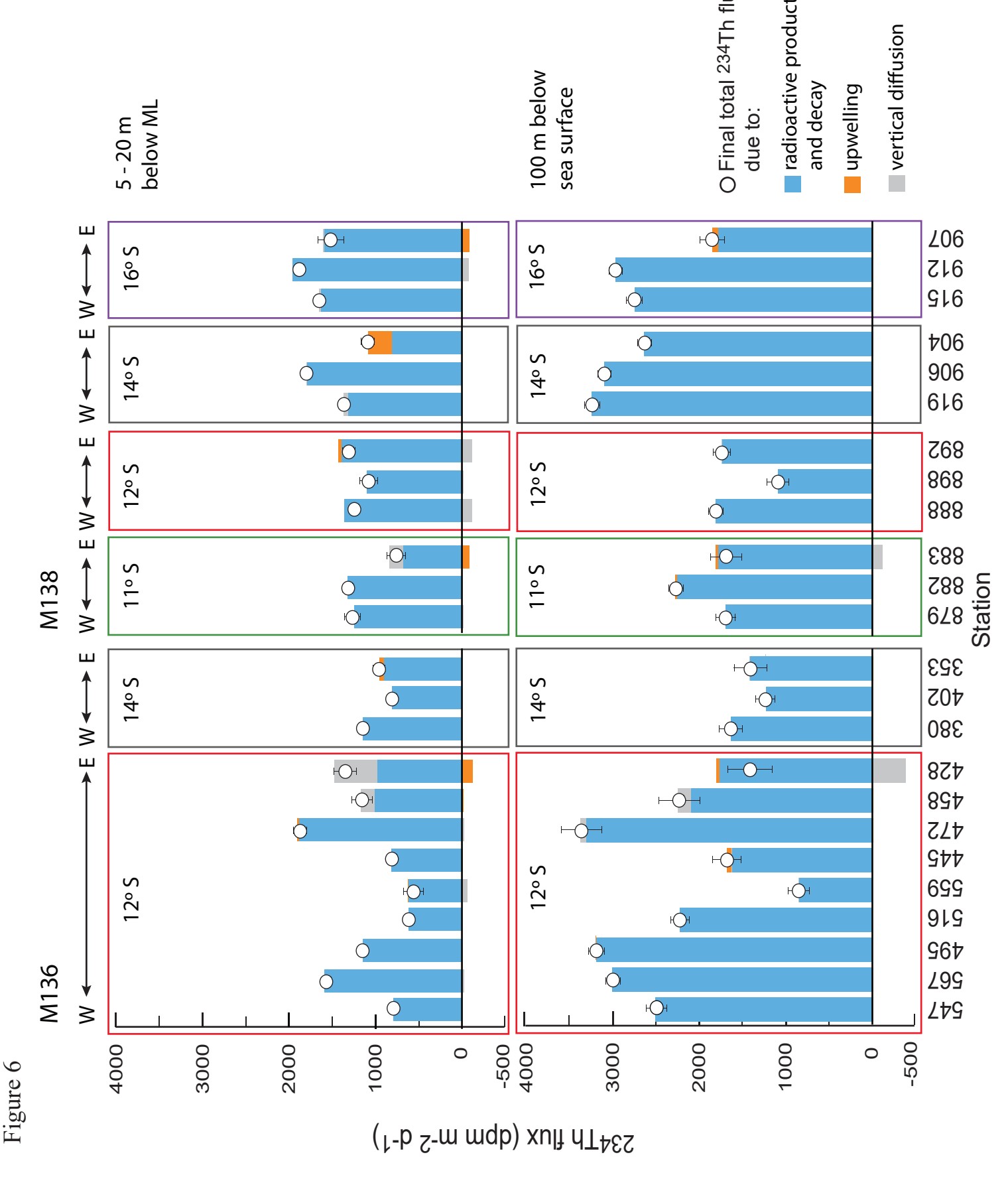

Figure 7

