# Peer review of "Effects of 238U variability and physical transport on water column"

_Biogeosciences, 2020_

## Referee Comment (RC1) · Anonymous Referee #1 · 7 Apr 2020

Xie et al. present new data of 234Th export fluxes from the coastal upwelling system off Peru, associated with an oxygen minimum zone (OMZ). The aim of this research is to investigate the effects of 238U variability and physical processes on the 234Th fluxes. The authors found a poor correlation between measured (by isotopic dilution) and calculated (from salinity) 238U activities. Even though only small variations were observed between measured and calculated 238U activities, this difference leads to significant underestimation of 234Th fluxes. 238U activities are usually not measured, as this represents additional work and the linear relationship with salinity is generally assumed. However, the current study clearly shows the need for measuring 238U activities in non-open ocean systems. The impact of physical processes, such as advection and diffusion, was evaluated by using ADCP, current velocities, satellite wind stress and in situ microstructure measurements. Unlike horizontal diffusion and advection, vertical diffusion and advection were found to significantly modify the 234Th export fluxes at shelf stations. Again, most studies neglect the impact of physics on 234Th fluxes and rare are those considering vertical/horizontal advection and diffusion effects on 234Th fluxes. Finally, the authors investigated the 234Th residence time and found a large temporal variation across the Peruvian upwelling zone, warning future studies to take into account these temporal changes while evaluating carbon export efficiencies. Overall, the manuscript is well written and represents an important effort. In most studies the influence of the 238U variability and physical processes are assumed to be negligible. The findings of this study are therefore highly valuable for the community. With some reorganisation and some more details on the calculations, this manuscript will be a good fit for publication in Biogeosciences.

1. Specific comments.

1-Results section. Details are missing to really understand the choices made in the Discussion. I propose to add a "Results" section that would moreover make the discussion clearer for the reader. This new section could present: 1) Total 234Th and 238U activities: basically what is written between the beginning of Section 3 and before the beginning of Section 3.1. 2) Export fluxes of 234Th: - Please give more details on the relevance of estimating fluxes at different horizon depths. First, clearly mention in the Methods that you calculate the export fluxes at 100m and below the mixed layer (ML). Then, in this new Results section, you could explain why you calculate the fluxes at 2 different depths. Why is it relevant to discuss fluxes at 100m or at the base of the ML for the purpose of this work? Also, explain why you estimate the fluxes "below" the ML and not simply at its base? - Steady state versus non-steady state (it should not be part of the sub-section dedicated to "dynamic advective and diffusive 234Th fluxes").

2-More details on the physical processes. Methods, section 2.3: For each physical process (horizontal advection, vertical advection, horizontal diffusion and vertical diffu-

[Figure]

sion), please give details on how the 234Th fluxes due to these processes are calculated. For example, lines 180-182, how do you use the daily wind stress to estimate the upwelling velocity? Lines 180-182 and lines 189-191: In addition to the cited references, please, briefly explain how VmADCP and in situ microstructure profiler measurements work and how you obtain current velocities or diffusivities from them?

Table 1: As not significant processes, you do not present the 234Th fluxes due to horizontal advection and diffusion. Please, give the values in Table 1 for comparison.

Discussion, section 3.2: Please, explain in more details how you calculate the vertical and horizontal 234Th gradients. Explanations about the vertical gradient are for example given by Black et al. (2018) and are useful for the reader. Moreover, lines 326-329, please, clearly say how you determine the horizontal 234Th gradient. What does "larger spatial scale" mean?

3-Greater 238U activities in suboxic environment. I am very surprised by these results as I would have expected the opposite, i.e., less U in anoxic/suboxic waters. This is very interesting and I did appreciate reading your possible explanations. I however have some questions about them: - Lines 265-267: If Fe reduction was going on, it would definitely be associated with U removal to the sediment. Is there enough U adsorbed on oxyhydroxides to outpace U removal? - Lines 270-273: Uranium enhancement related to flooding, strong rainfall and landslide would also come with freshwater. Don't you think this would also affect salinity? - Could an oxygenation event such as the one described by Rapp et al. (2020) during the 2015 El Niño be responsible of high U concentrations? Assuming a dynamic OMZ and assuming Uranium needs some time to equilibrate, would it be possible to measure high concentrations in low oxygen waters?

4-Residence times of 234Th. Lines 371-372: Please, explain how you estimate the residence times.

2. Line notes.

Line 97, line 104, line 113 and line 676: Keep similar wording all along the text and use "shelf-offshore transect" instead of "shelf-normal" or "shore-normal" transect.

Line 42: add "e.g." at the beginning of the citation list. There are many more studies.

Lines 119-120: Please, give the deep ocean average 234Th/238U ratio in your study.

Lines 167-168: Please, mention that fluxes are also estimated at 100m. You should also explain the reason of calculating fluxes at both 100m and at 5-20m below the ML for the relevance of this study – as justified for EZ and ML.

Line 183: "the depths correspond to 5-20m below the base of the ML", please mention that this is the reason why you calculate export fluxes at this horizon depth.

Line 203: would yield a maximum.., instead of would a yield maximum..

Lines 203-206: Maybe to move to the new "Results" section.

Lines 217-219: Please, provide the number of replicates for the IAPSO standard seawater: "$3.24 \pm 0.06$ ng/g, 1SD, n=?".

Lines: 249-254: "The consequence of this notable difference in 238U to 234Th flux according to Eq. (2) is neither linear nor straightforward, because the vertical gradients of both 238U and 234Th strongly affects the impacts of 238U variations on 234Th fluxes. In this study, 234Th fluxes at 100 m derived from S-based 238U lead to significant underestimation of 234Th fluxes by an average of 20% and as high as 40% (Table 2). These differences in 234Th fluxes will have direct consequences for 234Th derived elemental fluxes such as C, N, P and trace metals." This is a conclusion of the all section. I would thus move these sentences to the end of Section 3.1.

Line 260: S>12?

Lines 276-277: Shepherd et al., 2017 is not listed in the section "References".

Line 308: For comparison, please give also the fraction of upwelled 234Th fluxes compared to the total fluxes for the offshore stations.

Line 308: Cite Figure 5.

Line 322: Mean 234Th "activities" in the top layer.. ?

Line 322: Please precise what does "top layer" exactly mean. Like in the caption of Figure 6 and Table 3..

Line 355: cruises

Lines 380 and 382: Maybe change the 234Th activities into 234Th/238U ratios, as it might be easier to realise the magnitude of the deficit.

Lines 411-412: And 7Be isotopes, as you mention line 351.

Line 696: Error bars "are" (instead of were) indicated. Shelf instead of nearshore (to keep the same wording all along the manuscript).

Figure 2: It is difficult to see the small variations. Please, decrease the size of the 234Th data points and make the lines thinner. Add the error bars. If they are already indicated but too small to be seen, please mention it in the caption. The x axes are always the same, please, keep the O2 values only on the top of the figure and the 234Th, 238U, fluorescence values only on the bottom of the figure. By doing so, you can slightly increase the size of each graph. Please, keep your colour legend of Figure 1 and indicate the shelf to offshore transect by an arrow (maybe by writing W and E, like in Figure 5). Like in Black et al., 2018: indicate the depth of the mixed layer and the start of the Oxygen deficient zone.

Figure 3: Please indicate the error bars. If it is too much for the figures, I recommend to at least, indicate the size of the average error bar on a corner of the plot. Indicated the O2 concentrations in Figure 3c as well. This would confirm that the poor relationship does not depend on O2 concentrations.

Figure 4: There is no need to write the depths for each plot. Write the values only

on the left side of the figure. The legend has to be fixed and "fluorescence" has to be added on the bottom x axis of Figure 4c. In the legend of Figure 4a, define that the black dotted line corresponds to salinity and that the black solid line corresponds to temperature.

Figure 5: I do like this Figure: it is clear. Please modify the caption and write "5-20m below ML" instead of "base of the ML. In the legend, please write "Final total 234Th flux" for the white dots to keep the same wording than in Table 1.

Table 1: Please modify the caption and the top line of the 2nd column: "234Th flux 5-20m below the ML" instead of "below the ML" or "at the base of the ML".

References: Black et al., 2018. 234Th as a tracer of particulate export and remineralization in the southeastern tropical Pacific. Marine Chemistry. Rapp et al., 2020. El Niño-driven oxygenation impacts Peruvian shelf iron supply to the South Pacific Ocean. Geophysical Research Letters.

―――――――――――――――――――――――

---

## Referee Comment (RC2) · Anonymous Referee #2 · 16 Apr 2020

Major comments This study evaluated the impact from the non-linearity of U-S relationship, temporal variability of 234Th and 3-D physical transport of 234Th on the estimation of downward 234Th flux. I initially read the manuscript with interest but realized finally that I need to give it up. This is an important but difficult topic that has been ignored in various 234Th studies, while the superficial description and discussion on the data by the authors keep the manuscript from further acceptance. The non-linearity between 238U and salinity is interesting and I totally agree that will induce an over- or under-estimation on the final 234Th flux. I feel very nerves that the authors attributed such non-linearity to the flooding and landslides without any obvious evidences shown in the manuscript. Meanwhile, if it was true that high uranium was transported from

the coastal waters, then how was that for 234Th? I guess the 234Th activity could be low in the same water, and including the low 234Th water also elevated the 234Th flux calculation. The authors further examine the physical transport of 234Th, but again the in-depth discussion will be required. Quite a few descriptions and explanations should be listed here: The methods on the upwelling rate estimation using wind stress and its uncertainty, the diffusivity using in situ microstructure measurements and the detail calculation for horizontal advection (the equation 3 showed in the manuscript is way too simple for this paper). I strongly recommend the authors to add these parts in the methods and discussion during the revision, and most importantly, the evaluation of the uncertainty and error should be carefully done. For example, the authors calculated the upwelling rate was on the order of 10-6 to 10-7 m s-1, those values actually were quite low compared to other upwelling sites. In the last part of the discussion, the authors used a whole paragraph for the 234Th residence time. I did not find any wordings on the detailed calculation method for those residence time. I guess they are estimated using an 1-D steady state model, but given that the physical transport was important for some stations as the authors had pointed out, 3-D estimation for the 234Th residence time will also be needed. The 234Th and 238U data obtained in the region could be very interesting, the detailed description of their profiles should be more interesting. I think the authors should expand their methods part, and separate the result and discussion. In addition, I found some sentences in the conclusion should also move to the discussion. I also have quite a few detailed comments listed below. Minor comments: The title: Effects of 238U variability and physical transport. . . . . . . It gave me an impression that the author is evaluating the 238U transport which is actually 234Th. Page 3, Line 41, Add "in the upper ocean" after "export fluxes" Page 3, Line 47, Bhat et al., 1968 is not a appropriate reference, add some Santschi paper, and show the Kd values here. Page 3, Line 50-51, 234Th flux can be obtained even if you do not integrate with depth. Page 5, Methods part, Add the methods for the upwelling rate estimation, diffusivity calculation and current from ADCP. Page 6, Line 118-120, Did you just assume that 234Th had been in equilibrium with 238U or you

would acidify those sample and let them stay for a year until the equilibrium would be reached. Please make that clearer. Page 6, Line 125, 1 dpm or 10 dpm? Page 6, Line 125, what was the volume of your sample? 4L or 2L. Page 8, Line 171-172, Show the detailed calculation methods here or in the supplements. I guess here involved the simplification and manipulation of your data. Page 9, Line 180-181, I have concerned on the ADCP-data which are snapshots data during the cruise, while 234Th is a chemical tracer with a time integrated information included. How do you match the different time scale between the two parameters? Page 10, Line 208, Separation between results and discussion could be better. Page 11, Line 221-231, The detailed description of 234Th and 238U activities, ranges, averages, and their relationship with Chl a and oxygen will be appreciated. Page 13, Line 265-267, How about 234Th? Page 13, Line 268-273, This is too superficial? Do you have any optics data here? Page 14, Line 290-295, Show the equation for NSS calculation. I think in the supplement you will also need to explain how you do the error propagation. Page 14, Line 303, How reliable is your upwelling rate? I do not believe those numbers. Show the methods and put more discussion here. Page 15, Line 318, How much is "trivial"? less than 10Page 15, Line 325, How do you calculate the 234Th gradient? Page 16-17, Line 353-355, The time scale for the methods is very different. Page 17, Line 370, How do you do the calculation? 1D steady state? Or 3D steady State? Page 19, Line 411-414, not related, or move to discussion part. Page 19, Line 417-420, Move to discussion part? The references: all numbers of molecular weight for the isotopes should be in the upper case. There are quite a few errors on the references, please do the careful check. Figures: I think adding some figures here will be much helpful. Please add a transect distribution for 238U and 234Th to show the coast to offshore difference. And also add some profiles of the vertical diffusivity should be better. Figure 1: It is better to put the current field here in the map, or show it in a separate figure? Figure 2: Show the MLD and bottom depths here Figure 4, Can you show the profiles of 234Th for stations 458 and 508, although the surface sample was missing.

---

## Author Comment (AC1) · 14 May 2020

Xie et al. present new data of 234Th export fluxes from the coastal upwelling system off Peru, associated with an oxygen minimum zone (OMZ). The aim of this research is to investigate the effects of 238U variability and physical processes on the 234$^{Th}$ fluxes. The authors found a poor correlation between measured (by isotopic dilution) and calculated (from salinity) 238U activities. Even though only small variations were observed between measured and calculated 238U activities, this difference leads to significant underestimation of 234Th fluxes. 238U activities are usually not measured, as this represents additional work and the linear relationship with salinity is generally assumed. However, the current study clearly shows the need for measuring 238U activities in non-open ocean systems. The impact of physical processes, such as advection and diffusion, was evaluated by using ADCP, current velocities, satellite wind stress and in situ microstructure measurements. Unlike horizontal diffusion and advection, vertical diffusion and advection were found to significantly modify the 234Th export fluxes at shelf stations. Again, most studies neglect the impact of physics on 234Th fluxes and rare are those considering vertical/horizontal advection and diffusion effects on 234Th fluxes. Finally, the authors investigated the 234Th residence time and found a large temporal variation across the Peruvian upwelling zone, warning future studies to take into account these temporal changes while evaluating carbon export efficiencies. Overall, the manuscript is well written and represents an important effort. In most studies the influence of the 238U variability and physical processes are assumed to be negligible. The findings of this study are therefore highly valuable for the community. With some reorganisation and some more details on the calculations, this manuscript will be a good fit for publication in Biogeosciences.

**We thank the reviewer for his/her constructive comments. We've listed our point-by-point response in bold below.**

1. Specific comments.

1-Results section. Details are missing to really understand the choices made in the Discussion. I propose to add a "Results" section that would moreover make the discussion clearer for the reader. This new section could present: 1) Total 234Th and 238U activities: basically what is written between the beginning of Section 3 and before the beginning of Section 3.1. 2) Export fluxes of 234Th: - Please give more details on the relevance of estimating fluxes at different horizon depths. First, clearly mention in the Methods that you calculate the export fluxes at 100m and below the mixed layer (ML). Then, in this new Results section, you could explain why

you calculate the fluxes at 2 different depths. Why is it relevant to discuss fluxes at 100m or at the base of the ML for the purpose of this work? Also, explain why you estimate the fluxes "below" the ML and not simply at its base? - Steady state versus non-steady state (it should not be part of the sub-section dedicated to "dynamic advective and diffusive 234Th fluxes").

**Response: We have now added a new Results section to the manuscript that include details on the 234Th and 238U profiles, export of 234Th fluxes at 100 m and base of the ML (and why we used these two different depths), and comparison of the steady state versus non-steady state models. Please refer to the manuscript for more details.**

**Due to sampling logistics, we did not sample at the base of the ML, but 5-20 m below the ML. This depth corresponded closely to the EZ depth used in Black et al. (2018) in the same study area. For the purpose of comparison with earlier studies which reported 234Th fluxes at 100 m, we also calculated 234Th fluxes at 100 m in this study.**

2-More details on the physical processes. Methods, section 2.3: For each physical process (horizontal advection, vertical advection, horizontal diffusion and vertical diffusion), please give details on how the 234Th fluxes due to these processes are calculated. For example, lines 180-182, how do you use the daily wind stress to estimate the upwelling velocity? Lines 180-182 and lines 189-191: In addition to the cited references, please, briefly explain how VmADCP and in situ microstructure profiler measurements work and how you obtain current velocities or diffusivities from them?

**Response: We have now expanded the Methods section to include detailed descriptions of how upwelling rates, VmADCP -derived current velocities and microstructure-derived diffusivities were calculated.**

Table 1: As not significant processes, you do not present the 234Th fluxes due to horizontal advection and diffusion. Please, give the values in Table 1 for comparison.

**Response: We grouped stations within a 1° by 1° grid and calculated the average 234Th for the top layer, and large scale (1° apart) horizontal 234Th gradients were calculated based on this grouping. We then used average alongshore current velocities and eddy diffusivities for the flux estimation due to horizontal advection and diffusion. These estimations are correct to the first order. As these values were rough estimates, we feel that we should not include them in Table 1. We now explained these in more details in the manuscript.**

Discussion, section 3.2: Please, explain in more details how you calculate the vertical and horizontal 234Th gradients. Explanations about the vertical gradient are for example given by

Black et al. (2018) and are useful for the reader. Moreover, lines 326-329, please, clearly say how you determine the horizontal 234Th gradient. What does "larger spatial scale" mean?

**Response: We now specified these details on how we calculated the vertical and alongshore 234Th gradients in the Results section (new subsection 2.3.4).**

3-Greater 238U activities in suboxic environment. I am very surprised by these results as I would have expected the opposite, i.e., less U in anoxic/suboxic waters. This is very interesting and I did appreciate reading your possible explanations. I however have some questions about them:

- Lines 265-267: If Fe reduction was going on, it would definitely be associated with U removal to the sediment. Is there enough U adsorbed on oxyhydroxides to outpace U removal?

- Could an oxygenation event such as the one described by Rapp et al. (2020) during the 2015 El Niño be responsible of high U concentrations? Assuming a dynamic OMZ and assuming Uranium needs some time to equilibrate, would it be possible to measure high concentrations in low oxygen waters?

**Response to both comments: The presence of high content of organic matter in the Peruvian sediments greatly influence U mobility and promote U sorption onto mineral surfaces, such as Fe hydroxides. However, the reviewer is correct that U reduction and removal should occur when sedimentary Fe reduction took place. This was indeed what was observed on the Peruvian shelf by an earlier study (Scholz et al. 2011). We now significantly toned down the discussion on Fe reduction being the main additional U source to the water column, as we cannot accurately quantify the amount of remobilized adsorbed-U vs. U removal.**

**The same study by Scholz et al. (2011) further showed considerable diffusive U fluxes out of the sediments along the Peru shelf where both Fe reduction and U reduction took place. This remobilization of U was attributed to ENSO-related transient U re-oxidation and recycling. It was suggested that a minute increase in bottom water oxygen concentration was sufficient to shift the U(VI)/U(IV) boundary by a few centimeters and remobilize authigenic U. The coastal El Niño developed preceding to and during our sampling campaign could induce an oxygenation event large enough to remobilized authigenic U along the Peruvian shelf. We now added this discussion to our manuscript.**

**We would also like to point out that we have significantly modified the discussion on the U-salinity relationship. We now acknowledged that poor U-salinity correlations were also observed in other open ocean basins, and explored possible explanations for this poor correlation in our study area.**

- Lines 270-273: Uranium enhancement related to flooding, strong rainfall and landslide would also come with freshwater. Don't you think this would also affect salinity?

**Response: This is a fair point. Flooding likely affected both U and salinity in coastal waters. The addition of freshwater and riverine U may draw the datapoints up and down the conservative mixing line (as shown in Owens et al. 2011). However, this was not the case in our study where majority of the U data points fall above the S-U line defined by Owens et al. (2011), indicative of other governing processes other than conservative mixing. We have disregarded the discussion of coastal flooding being one of the main causes of the poor U-salinity correlation.**

4-Residence times of 234Th. Lines 371-372: Please, explain how you estimate the residence times.

**Response: We now specified in the Methods section the formulation and details on how we calculated the residence times.**

2. Line notes.

Line 97, line 104, line 113 and line 676: Keep similar wording all along the text and use "shelf-offshore transect" instead of "shelf-normal" or "shore-normal" transect.

**Response: We now used the wording "shelf-offshore transect" instead of the technical term "shore-normal transect".**

Line 42: add "e.g." at the beginning of the citation list. There are many more studies.

**Response: fixed**

Lines 119-120: Please, give the deep ocean average 234Th/238U ratio in your study.

**Response: We now added this average 234Th/238U ratio in the Results section**

Lines 167-168: Please, mention that fluxes are also estimated at 100m. You should also explain the reason of calculating fluxes at both 100m and at 5-20m below the ML for the relevance of this study – as justified for EZ and ML.

**Response: We now specified in the Results section why we chose two different depths for the flux calculation.**

Line 183: "the depths correspond to 5-20m below the base of the ML", please mention that this is the reason why you calculate export fluxes at this horizon depth.

**Response: fixed**

Line 203: would yield a maximum.., instead of would a yield maximum..

**Response: fixed**

Lines 203-206: Maybe to move to the new "Results" section.

**Response: fixed**

Lines 217-219: Please, provide the number of replicates for the IAPSO standard seawater: "3.24 ± 0.06 ng/g, 1SD, n=?".

**Response: fixed**

Lines: 249-254: "The consequence of this notable difference in 238U to 234Th flux according to Eq. (2) is neither linear nor straightforward, because the vertical gradients of both 238U and 234Th strongly affects the impacts of 238U variations on 234Th fluxes. In this study, 234Th fluxes at 100 m derived from S-based 238U lead to significant underestimation of 234Th fluxes by an average of 20% and as high as 40% (Table 2). These differences in 234Th fluxes will have direct consequences for 234$^{Th}$ derived elemental fluxes such as C, N, P and trace metals." This is a conclusion of the all section. I would thus move these sentences to the end of Section 3.1.

**Response: fixed**

Line 260: S>12?

**Response: It should be S > 10 as stated in the original text. McKee et al. (1987) and Swarzenski et al. (2004) looked at two different salinity thresholds in terms of the 238U-S linear correlation.**

Lines 276-277: Shepherd et al., 2017 is not listed in the section "References".

**Response: fixed**

Line 308: For comparison, please give also the fraction of upwelled 234Th fluxes compared to the total fluxes for the offshore stations.

**Response: The fraction of upwelled 234Th fluxes compared to total fluxes is now quoted in text.**

Line 308: Cite Figure 5.

**Response: quoted**

Line 322: Mean 234Th "activities" in the top layer.. ?

**Response: fixed**

Line 322: Please precise what does "top layer" exactly mean. Like in the caption of Figure 6 and Table 3..

**Response: fixed**

Line 355: cruises

**Response: fixed**

Lines 380 and 382: Maybe change the 234Th activities into 234Th/238U ratios, as it might be easier to realise the magnitude of the deficit.

**Response: The reviewer provided a very good suggestion here, but we unfortunately cannot proceed for one important reason: the activities of 238U in Black et al. (2017) were not measured nor are they reported in the available GEOTRACES database.**

Lines 411-412: And 7Be isotopes, as you mention line 351.

**Response: added**

Line 696: Error bars "are" (instead of were) indicated. Shelf instead of nearshore (to keep the same wording all along the manuscript).

**Response: fixed**

Figure 2: It is difficult to see the small variations. Please, decrease the size of the 234Th data points and make the lines thinner. Add the error bars. If they are already indicated but too small to be seen, please mention it in the caption. The x axes are always the same, please, keep the O2 values only on the top of the figure and the 234Th, 238U, fluorescence values only on the bottom of the figure. By doing so, you can slightly increase the size of each graph. Please, keep your colour legend of Figure 1 and indicate the shelf to offshore transect by an arrow (maybe by writing W and E, like in Figure 5). Like in Black et al., 2018: indicate the depth of the mixed layer and the start of the Oxygen deficient zone.

**Response: Figure 2 was modified according to most of these comments. The depth of the mixed layer is now indicated by a red dashed line. The start of the oxygen deficient zone is where oxygen diminishes. We did not use color legend from Figure 1 to keep Figure 2 clean and easy to read.**

Figure 3: Please indicate the error bars. If it is too much for the figures, I recommend to at least, indicate the size of the average error bar on a corner of the plot. Indicated the O2 concentrations in Figure 3c as well. This would confirm that the poor relationship does not depend on O2 concentrations.

**Response: We now added error bars, which are smaller than symbols. We also added oxygen concentrations in Figure 3c.**

Figure 4: There is no need to write the depths for each plot. Write the values only on the left side of the figure. The legend has to be fixed and "fluorescence" has to be added on the bottom x axis of Figure 4c. In the legend of Figure 4a, define that the black dotted line corresponds to salinity and that the black solid line corresponds to temperature.

**Response: We fixed the vertical axes and Figure 4c horizontal axis (now Figure 5). It is not correct regarding reviewer's comment on the dashed and solid lines. We specified in the caption that the dashed lines corresponded to temperature and dashed lines corresponded to salinity.**

Figure 5: I do like this Figure: it is clear. Please modify the caption and write "5-20m below ML" instead of "base of the ML. In the legend, please write "Final total 234$^{Th}$ flux" for the white dots to keep the same wording than in Table 1.

**Response: Fixed**

Table 1: Please modify the caption and the top line of the 2nd column: "234Th flux 5-20m below the ML" instead of "below the ML" or "at the base of the ML".

**Response: Fixed**

---

## Author Comment (AC2) · 14 May 2020

**We thank the reviewer for his/her constructive comments. We've listed our point-by-point response in bold below.**

Major comments This study evaluated the impact from the non-linearity of U-S relationship, temporal variability of 234Th and 3-D physical transport of 234Th on the estimation of downward 234Th flux. I initially read the manuscript with interest but realized finally that I need to give it up. This is an important but difficult topic that has been ignored in various 234Th studies, while the superficial description and discussion on the data by the authors keep the manuscript from further acceptance. The non-linearity between 238U and salinity is interesting and I totally agree that will induce an over- or under-estimation on the final 234Th flux. I feel very nerves that the authors attributed such non-linearity to the flooding and landslides without any obvious evidences shown in the manuscript. Meanwhile, if it was true that high uranium was transported from the coastal waters, then how was that for 234Th? I guess the

234Th activity could be low in the same water, and including the low 234Th water also elevated the 234Th flux calculation.

**Response: The coastal El Niño of 2017 induced coastal precipitation as strong as the 1997-98 El Niño (Echevin et al. 2008), resulting in devastating flooding and landslides in central and**

**northern coastal Peru. Evidence of this coastal El Niño has been presented in earlier studies cited in our manuscript. This intense flooding likely delivered large amount of fresh water, dissolved and particulate 238U, and possibly particulate 234Th. 234Th is highly particle reactive so it is unlikely that the coastal flooding has directly introduced dissolved 234Th to the coastal water. 234Th produced in-situ within the upper water column was likely**

**scavenged quickly due to enhanced particulate input from land at this time. The addition of freshwater and riverine U may draw the datapoints up and down the conservative mixing line (as shown in Owens et al. 2011). However, this was not the case in our study where majority of the U data points fall above the S-U line defined by Owens et al. (2011). We thus agreed with the reviewer that coastal flooding is unlikely the cause of such deviations. We now have**

**significantly modified the discussion regarding the non-linear U-salinity correlation based on both reviewers' comments to include U remobilization induced by bottom water oxygenation**

**being one of the main mechanisms for enhanced water column U. We have disregarded the discussion of coastal flooding being one of the main causes of the poor U-salinity correlation.**

**Echevin, V. M., Colas, F., Espinoza-Morriberon, D., Anculle, T., Vasquez, L., and Gutierrez, D.:**
**Forcings and evolution of the 2017 coastal El Niño off Northern Peru and Ecuador, Frontiers in Marine Science, 5, 367, https://doi.org/10.3389/fmars.2018.00367, 2018.**

**Owens, S., Buesseler, K., and Sims, K.: Re-evaluating the 238U-salinity relationship in seawater: Implications for the 238U–234Th disequilibrium method, Marine Chemistry, 127, 31-39, https://doi.org/10.1016/j.marchem.2011.07.005, 2011.**

The authors further examine the physical transport of 234Th, but again the in-depth discussion will be required. Quite a few descriptions and explanations should be listed here: The methods on the upwelling rate estimation using wind stress and its uncertainty, the diffusivity using in situ microstructure measurements and the detail calculation for horizontal advection (the
equation 3 showed in the manuscript is way too simple for this paper). I strongly recommend the authors to add these parts in the methods and discussion during the revision, and most importantly, the evaluation of the uncertainty and error should be carefully done. For example, the authors calculated the upwelling rate was on the order of 10-6 to 10-7 m s-1, those values actually were quite low compared to other upwelling sites.

**Response: We have expanded the Methods section to include essential details of how upwelling rates, current velocities and diffusivities were estimated. We also include methods for error propagation in the Supplement. In the Results section, we detailed 234Th fluxes due to radioactive production and decay, advection and diffusion.**

**Upwelling rates off Peru estimated in our study were indeed smaller than some of the**
**upwelling rates in other upwelling sites, but is in accord with the atmospheric and oceanic conditions off Peru at the time of sample collection. Wind stress were unusually weak off Peru beginning the last quarter of 2016 and lasted until the first half of March 2017. Toward the end of March 2017, an increase of the nearshore wind stress and a relaxation in offshore wind stress off northern Peru generated an intense wind stress curl anomaly and an**
**associated downwelling (e.g., Echevin et al. 2008). An SST transect along 12°S off Peru showed that upwelling was restricted to the shelf and in the upper 50 m. These atmospheric and oceanic conditions were unique and resulted in very weak upwelling rates off Peru.**

**Lüdke, J., et al. (in review 2019). "Influence of intraseasonal eastern boundary circulation**
**variability on hydrography and biogeochemistry off Peru." Ocean Sci. Discuss. 2019: 1-31.**

In the last part of the discussion, the authors used a whole paragraph for the 234Th residence time. I did not find any wordings on the detailed calculation method for those residence time. I guess they are estimated using an 1-D steady state model, but given that the physical transport was important for some stations as the authors had pointed out, 3-D estimation for the 234Th residence time will also be needed.

**Response: We now included the formulation for estimation of residence time, which was based on a 1D steady state model. Although this 1D steady state model is an oversimplification of a multi-dimensional process and should be used with caution, it provides a good first order estimate for understanding the highly dynamic nature of the 234Th residence time. It also provides a reasonable value that can be directly compared to values estimated in earlier 234Th flux studies that did not consider physical processes. We now added this discussion to the main text.**

The 234Th and 238U data obtained in the region could be very interesting, the detailed description of their profiles should be more interesting.

**Response: We now included a Result section that describe both 234Th and 238U profiles in detail.**

I think the authors should expand their methods part, and separate the result and discussion. In addition, I found some sentences in the conclusion should also move to the discussion.

**Response: We now expanded the Methods part to include detail description on how upwelling rates, current velocities and diffusivities were calculated. We also separated the Results and Discussion.**

I also have quite a few detailed comments listed below. Minor comments:

The title: Effects of 238U variability and physical transport. . .. . .. It gave me an impression that the author is evaluating the 238U transport which is actually 234Th.

**Response: The manuscript looks into the roles of 238U variability and physical transport on 234Th distribution and transport, and we think that the title fully reflects the goals and findings of this manuscript.**

Page 3, Line 41, Add "in the upper ocean" after "export fluxes"

**Response: fixed**

Page 3, Line 47, Bhat et al., 1968 is not a appropriate reference, add some Santschi paper, and show the Kd values here.

**Response: We disagree with the reviewer. Bhat et al. (1968) is one of the earlier field studies**
**that have demonstrated the particle reactive nature of 234Th in the ocean.  We now added**
**the Kd values with reference to Santschi et al. (2006).**

Page 3, Line 50-51, 234Th flux can be obtained even if you do not integrate with depth.

**Response: It is necessary to integrate $^{234}$Th activities with depth in order to estimate $^{234}$Th**
**flux.**

Page 5, Methods part, Add the methods for the upwelling rate estimation, diffusivity calculation
and current from ADCP.

**Response: We added methods on how upwelling rates, current velocities and diffusivities**
**were calculated**

Page 6, Line 118-120, Did you just assume that 234Th had been in equilibrium with 238U or you
would acidify those sample and let them stay for a year until the equilibrium would be reached.
Please make that clearer.

**Response: Only 238U samples were acidified. We now clarified this in the manuscript.**

Page 6, Line 125, 1 dpm or 10 dpm?

**Response: It is 1 dpm as stated in the text.**

Page 6, Line 125, what was the volume of your sample? 4L or 2L.

**Response: We now specified 4L as the sample volume.**

Page 8, Line 171-172, Show the detailed calculation methods here or in the supplements. I
guess here involved the simplification and manipulation of your data.

**Response: We now added details in the Methods section on how upwelling velocities, current velocities, and diffusivities were calculated. We also added details on how vertical and horizontal gradients were calculated.**

Page 9, Line 180-181, I have concerned on the ADCP-data which are snapshots data during the
cruise, while 234Th is a chemical tracer with a time integrated information included. How do you match the different time scale between the two parameters?

**Response: We appreciate the reviewer's concerned. As stated in Lines 186-189 in the original manuscript, "Zonal and meridional current velocities for each station were averaged over 5 days before and after station occupation. These current velocities were further averaged over**
**a 10 km radian at stations closest to shore (St. 353, 428, 458, 475, 508, 904, and 907) and over a 50 km radian at the rest of the stations." In another word, the ADCP-derived current velocities were averaged over a 10-day timescale. This timescale is somewhat shorter than the residence time of 234Th. But given the short cruise timeframe, we consider this time averaged appropriate.**

Page 10, Line 208, Separation between results and discussion could be better.

**Response: We now separated results and discussion.**

Page 11, Line 221-231, The detailed description of 234Th and 238U activities, ranges, averages,
and their relationship with Chl a and oxygen will be appreciated.

**Response: We added detailed descriptions in the new Results section.**

Page 13, Line 265-267, How about 234Th?

**Response: Unlike U, Th is not redox sensitive.**

Page 13, Line 268-273, This is too superficial? Do you have any optics data here?

**Response: Numerous evidence of the 2017 coastal El Niño off Peru has been published in previous studies, which were referenced in our manuscript.**

**Here we referenced to a figure by Echevin et al. (2018) who showed that the magnitude of**
**precipitation in the eastern equatorial Pacific during the 2017 coastal El Niño was almost as intense as that during the 1997-98 El Niño event:**

[Figure]

FIGURE 1 | (A) SST anomaly (in °C) and (B) SST anomaly (in °C, color scale) and precipitation (in mm day⁻¹, contours) during the 1997–1998 El Niño in March 1998; (C,D) are same as (A,B) during the 2017 coastal El Niño in March 2017. Anomalies were computed with respect to the 1981–2016 climatology.

Page 14, Line 290-295, Show the equation for NSS calculation. I think in the supplement you will also need to explain how you do the error propagation.

**Response: We now referenced readers to Resplandy et al. (2012) and Savoye et al. (2006) for details regarding the derivation of NSS flux formulation and error propagation.**

Page 14, Line 303, How reliable is your upwelling rate? I do not believe those numbers. Show the methods and put more discussion here.

**Response: We now showed details in the Methods section how we calculated the upwelling rates. Please also refer to our response above in Line 53-58 in this document, which we showed that the upwelling rates estimated in our study were reliable.**

Page 15, Line 318, How much is "trivial"? less than 10

**Response: We now specified it as "insignificant, ranging between 1% and 10%" instead of "trivial".**

Page 15, Line 325, How do you calculate the 234Th gradient?

**Response: We grouped stations within a 1° by 1° grid and calculated the average 234Th for the top layer, and large scale (1° apart) horizontal 234Th gradients were calculated based on this grouping. We now added details in the Results section on how vertical and horizontal gradients were calculated.**

Page 16-17, Line 353-355, The time scale for the methods is very different.

     **Response: Agreed. We now specified these two methods estimate upwelling rates at different timescales.**

     Page 17, Line 370, How do you do the calculation? 1D steady state? Or 3D steady State?

**Response: Please refer to our response in Line 70-76 in this document.**

     Page 19, Line 411-414, not related, or move to discussion part.

     Page 19, Line 417-420, Move to discussion part?

     **Response to both comments: Both are relevant in terms of implications for future coastal**
**234Th flux studies.**

     The references: all numbers of molecular weight for the isotopes should be in the upper case. There are quite a few errors on the references, please do the careful check.

     **Response: fixed**

     Figures: I think adding some figures here will be much helpful. Please add a transect distribution for 238U and 234Th to show the coast to offshore difference. And also add some profiles of the vertical diffusivity should be better.

     **Response: We now added a figure of 234Th/238U transects to show the distributions of shelf-**
**offshore 234Th deficits (as Figure 3 in the revised manuscript). Diffusivity profiles were shown in the original supplementary file.**

     Figure 1: It is better to put the current field here in the map, or show it in a separate figure?

     **Response: We now added the current field in Figure 1.**

Figure 2: Show the MLD and bottom depths here

**Response: We now indicated the MLD for all stations and bottom depths for stations whose bottom depths are shallower than 600 m (scale of y-axis).**

Figure 4, Can you show the profiles of 234Th for stations 458 and 508, although the surface sample was missing.

**Response: We now showed the comparison between stations 458 and 508 in Figure 4 (Figure 6 in the revised manuscript).**

---

## Referee Report (RR1)

I enjoyed reading this new version of the manuscript: it is now well-composed and the importance of physical processes and 238U variability on the estimation of downward 234Th export fluxes is clear. I recommend acceptance following minor revisions.

- Introduction

Line 43. There are many more studies investigating the elemental export fluxes. Please add "e.g." [Bhat et al., 1968, etc.]

Line 50. and "is" [thus strongly scavenged..]

Line 55. You can also add Si to the list C, N, P, trace metals.

Line 55. I also would add "e.g." [Bhat et al., 1968, etc..] as it is a succinct list of studies investigating elemental export fluxes.

Lines 74-75: I think this paragraph break is not necessary. Both paragraphs speak about advection and diffusion effect on 234Th fluxes.

Line 75. I think you can delete "that" and add "to" [be included..]

- Methods

Lines 140-147: Please mention why Mn would be a problem during the ICP MS analyses of 229Th /230Th ratios.

Line 149: Why do you use 1N HNO3? Usually ICPMS analyses are made with 2% HNO3 (i.e. 0.3N).

Lines 164-166: The recommended time interval between two visits is of >2 weeks. What would a time interval of maximum 4.5 days imply for your study?

Lines 175-185: I understand the depth for estimating the export fluxes is only of little relevance, but please, indicate at which depths you estimated the fluxes here and why these depths: 100m (for comparing with other studies) and 5-20m below the ML (not exact ML because of sampling logistics).

Lines 205-207: Please give the values of the Coriolis parameter ($f$) and the water density ($\rho$) you used and from where they come from.

Line 223: Please, precise how you estimate the upwelling rate from the vertical velocity ($w$)? By interpolation between 0 and 240m?

Lines 236-237: Please mention where these velocities can be found. In Lüdke et al., in review?

Lines 246-247: Which other cruises are you referencing?

Line 251: "At most CTD stations": for which stations do you not have microstructure profiles?

Line 255: Which value did you use for the stratification ($N$) and from where does it come from?

Line 277: Why do you use $\tau_{1/2}$? Please remove ½ if not needed.

Line 284: Please precise the surface layers are until 30m for M136 and 50m for M138, and explain why these depths.

- Results

Line 305: Total 234Th "activities"

Line 310: Please cite Table 1, where equilibrium depths are showed.

Line 326: Why do you use top layers as top 30m and top 50m? If you agree with one of my previous comment this will be explained in the Methods (line 284).

Line 397: Within the *upper* (?) 27 and 33m layer at *offshore* (?) deep stations

Line 415: please provide again the depths of the surface layer.

Lines 420-421: I think this paragraph break is not necessary. Both paragraphs speak about horizontal advection and diffusion effect on 234Th fluxes.

- Discussion

Line 443: Delete recent (it was 14 and 9 years ago already!)

Line 446: "a minute increase" is not clear. Do you mean that even a short-term oxygenation event, of the order of the minute, could release U from the sediments? If it is the case, please re write.

Lines 461-479: The explanation is not yet clear. Understanding how U (or Fe) reduction and remobilization could occur "at the same time" was not straightforward at first reading. Please, make a clear distinction between the ocean-sediment interface where $O_2$ concentrations can increase, realising U; and the suboxic/anoxic sediments where U is reduced and trapped.
Is there a study you could cite to support that a strong El Nino event (such as the one preceding your cruise) could induce an oxygenation event large enough to release U (lines 472-473)?

Line 503: In order to easily compare with GP16, please give the values you estimated here.

Line 505: same comment: please say how much were upwelling fluxes accounting for in your study.

Figure 2: Please add in both the caption and legend what the red line corresponds to.

---

## Referee Report (RR2)

**Second round comments on "Effects of $^{238}$U variability and physical transport on the water column $^{234}$Th downward fluxes in the coastal upwelling system off Peru" by Xie et al.**

**Anonymous reviewer #2**

I am very happy to see this version of the manuscript which I read with more fun than their first version. Most of my concerns has been answered in the new version. There are not too many studies trying to discuss the impact of physical transport on the downward $^{234}$Th flux in the open ocean. This study is therefore welcome to the community.

Before the acceptance of this paper, I only have one question that I am not satisfied. The authors attributed the abnormal uranium activity to the flooding from the coasts, and they also indicated a high activity of particulate $^{234}$Th in those flooding waters. Therefore, the dissolved $^{234}$Th in those water should be lowered by the sinking of those riverine particles. Once the water was transferred to the region of sampling, it should represent an integrated signal mostly derived from flooding particle export not just the local marine particle export. Then even we have carefully estimated the horizonal and vertical transport of $^{234}$Th, the final $^{234}$Th flux is still not induced by the local export. I do not know for this case $^{234}$Th is still a good tracer or not?

---

## Author Response (AR2)

**We thank both reviewers again for their constructive comments. Our point-by-point response to their comments is highlighted in bold in this document.**

Reviewer #1

I enjoyed reading this new version of the manuscript: it is now well-composed and the importance of physical processes and 238U variability on the estimation of downward 234Th export fluxes is clear. I recommend acceptance following minor revisions.

☐ Introduction

Line 43. There are many more studies investigating the elemental export fluxes. Please add "e.g." [Bhat et al., 1968, etc.]

**Response: fixed**

Line 50. and "is" [thus strongly scavenged..]

**Response: fixed**

Line 55. You can also add Si to the list C, N, P, trace metals.

**Response: fixed**

Line 55. I also would add "e.g." [Bhat et al., 1968, etc..] as it is a succinct list of studies investigating elemental export fluxes.

**Response: fixed**

Lines 74-75: I think this paragraph break is not necessary. Both paragraphs speak about advection and diffusion effect on 234Th fluxes.

**Response: We consider the paragraph break appropriate, as the previous paragraph discussed how the single box 234Th models are inappropriate in parts of the open ocean and the second paragraph discussed how 234Th fluxes in coastal regions are more vulnerable to physical processes. This paragraph break is also necessary to avoid a super long paragraph.**

Line 75. I think you can delete "that" and add "to" [be included..]

**Response: For clarity, we modified the sentence as "The dynamic nature of coastal processes requires that physical terms should be included in 234Th flux calculation whenever possible."**

☐ Methods

Lines 140-147: Please mention why Mn would be a problem during the ICP MS analyses of 229Th /230Th ratios.

**Response: Mn was not a problem for 229Th /230Th ratios but for the maintenance of ICPMS itself. The large amount of Mn in the samples means that the ICPMS needs to be cleaned thoroughly for days after each 229Th /230Th session before other trace-metal users could use it again. Performing column chemistry to remove Mn in 234Th samples before ICPMS analysis is now an agreement among lab users in our group.**

Line 149: Why do you use 1N HNO3? Usually ICPMS analyses are made with 2% HNO3 (i.e. 0.3N).

**Response: It is not uncommon to dilute samples in 1N HNO3 for ICPMS analysis. Typical ICPMS standard solutions are in 2% (0.32N) - 5% (0.8N) HNO3 solutions, and a few are in 10% (1.6N) HNO3 solutions. Our lab has been using 1N HNO3 for ICPMS analyses, which has been working perfectly well for various trace elements.**

Lines 164-166: The recommended time interval between two visits is of >2 weeks. What would a time interval of maximum 4.5 days imply for your study?

**Response: We modified line 362-364 in Results to address this comment: "The large errors associated with the non-steady state calculation due to the short duration between station occupations prevent a meaningful application of this model in the current study (also see discussion in Resplandy et al, 2012)."**

Lines 175-185: I understand the depth for estimating the export fluxes is only of little relevance, but please, indicate at which depths you estimated the fluxes here and why these depths: 100m (for comparing with other studies) and 5-20m below the ML (not exact ML because of sampling logistics).

**Response: The reasoning on at which depths fluxes were calculated were originally given in lines 372-376 (Results section): "Due to sampling logistics, we did not sample at the base of the ML, but 5-20 m below the ML. This depth corresponded closely to the EZ depth used in Black et al. (2018) in the same study area during austral spring 2013. For the purpose of comparison with earlier studies which reported 234Th fluxes at 100 m, we also calculated 234Th fluxes at 100 m in this study." We now moved this paragraph to Line 185.**

Lines 205-207: Please give the values of the Coriolis parameter ($f$) and the water density ($\rho$) you used and from where they come from.

**Response: The Coriolis parameter is not a fixed value but a function of latitude. We now specified the water density in the text.**

Line 223: Please, precise how you estimate the upwelling rate from the vertical velocity ($w$)? By interpolation between 0 and 240m?

**Response: The reviewer is correct that upwelling rate was interpolated between 0 and 240 m. We also stated in the original text that "We assumed a linear decrease of w from base of the mixed layer toward both the ocean surface and 240 m depth (bottom depth of our shallowest station). Upwelling rates at any depth between 0 and 240 m at individual stations could thus be determined once w was estimated."**

Lines 236-237: Please mention where these velocities can be found. In Lüdke et al., in review?

**Response: We now referenced (Lüdke et al., in review 2020).**

Lines 246-247: Which other cruises are you referencing?

**Response: We now specified M137 as the follow-up cruise.**

Line 251: "At most CTD stations": for which stations do you not have microstructure profiles?

**Response: For clarification, we modified this sentence to state "On transit between each CTD station 3 to 9 microstructure profiles were collected". Pleas also note that we stated in**
**Line 261, "An average turbulent vertical diffusivity profile was calculated from all inshore (<500m water depth) and from all offshore (>500m water depth) profiles (Figure S1)". So we would like to reiterate that inshore and offshore vertical diffusivity profiles are not individual profiles, but averages of all relevant microstructure profiles in proximation to our CTD stations during cruise M136 and M137.**

Line 255: Which value did you use for the stratification ($N$) and from where does it come from?

**Response: Stratification (Buoyancy frequency) was calculated using CTD data retrieved from microstructure profilers and following the gsw_Nsquared function from the Gibbs**
**Sea Water library (McDougall et al., 2009; Roquet et al., 2015). A running mean of 10 dbar was applied to avoid including unstable events due to turbulent overturns. We now added this discussion to the text.**

Line 277: Why do you use $\tau$ 1/2? Please remove ½ if not needed.

**Response: We now use $\tau_{Th}$ as 234Th residence time to avoid confusion with $\tau$ (wind stress)**

Line 284: Please precise the surface layers are until 30m for M136 and 50m for M138, and explain why these depths.

**Response: To avoid repetition throughout the text, we now added in Line 243: "As representative for the near-surface flow, we extracted the velocity data from the top 30 m for M136 stations and top 50 m for M138 station (defined as the "top layer" thereafter); these depths correspond to 5-20 m below the base of the ML during each cruise." Note that**
**we've opted to use the term "top layer" throughout the text instead of "surface layer" to avoid confusion.**

Results

Line 305: Total 234Th "activities"

       **Response: fixed**

       Line 310: Please cite Table 1, where equilibrium depths are showed.

       **Response: fixed**

       Line 326: Why do you use top layers as top 30m and top 50m? If you agree with one of my
       previous comment this will be explained in the Methods (line 284).

       **Response: Please refer to our replies in Lines 73 and 128 in this document.**

       Line 397: Within the *upper* (?) 27 and 33m layer at *offshore* (?) deep stations

**Response: fixed**

       Line 415: please provide again the depths of the surface layer.

       **Response: Please refer to our reply in Line 128 in this document.**

       Lines 420-421: I think this paragraph break is not necessary. Both paragraphs speak about
       horizontal advection and diffusion effect on 234Th fluxes.

       **Response: The former paragraph discussed advection fluxes and the latter one discussed**

**diffusive fluxes. It would be better to have the paragraph break.**

        Discussion

       Line 443: Delete recent (it was 14 and 9 years ago already!)

       **Response: fixed**

       Line 446: "a minute increase" is not clear. Do you mean that even a short-term oxygenation
       event, of the order of the minute, could release U from the sediments? If it is the case, please re write.

       **Response: "a minute increase" means an increase of oxygen of an extremely tiny amount,**
       **often below detection limits. We feel that the wording "minute" best described in a**
       **scientific manner such extremely small change in oxygen concentrations.**

       Lines 461-479: The explanation is not yet clear. Understanding how U (or Fe) reduction and
       remobilization could occur "at the same time" was not straightforward at first reading. Please, make a clear distinction between the ocean-sediment interface where O2 concentrations can increase, realising U; and the suboxic/anoxic sediments where U is reduced and trapped.

Is there a study you could cite to support that a strong El Nino event (such as the one preceding your cruise) could induce an oxygenation event large enough to release U (lines 472-473)?

**Response: We agreed with the reviewer and have now added the following discussion to Line 468: "In reducing pore water, U reduction and removal from pore water is usually**

**seen within the Fe reduction zone (Barnes and Cochran, 1990; Barnes and Cochran, 1991; Scholz et al., 2011). As such, a downward diffusive flux of U across the water-sediment interface is expected in reducing sedimentary environment." A reference to Scholtz et al. (GCA, 2011) was included in the original text which showed clear evidence that a tiny increase in bottom water oxygen concentrations would be sufficient to release U.**

Line 503: In order to easily compare with GP16, please give the values you estimated here.

**Response: Agreed and fixed**

Line 505: same comment: please say how much were upwelling fluxes accounting for in your study.

**Response: We now specified how much upwelling fluxes made up the total 234Th fluxes in our study.**

Figure 2: Please add in both the caption and legend what the red line corresponds to.

**Response: The caption in the original text stated "Red dashed lines indicate the depth of the mixed layer." For clarity, we only included data symbols in the legend.**

Reviewer #2
Second round comments on "Effects of 238U variability and physical transport on the water column 234Th downward fluxes in the coastal upwelling system off Peru" by Xie et al.

Anonymous reviewer #2

I am very happy to see this version of the manuscript which I read with more fun than their first
version. Most of my concerns has been answered in the new version. There are not too many studies trying to discuss the impact of physical transport on the downward 234Th flux in the open ocean. This study is therefore welcome to the community.

Before the acceptance of this paper, I only have one question that I am not satisfied. The authors attributed the abnormal uranium activity to the flooding from the coasts, and they also indicated
a high activity of particulate 234Th in those flooding waters. Therefore, the dissolved 234Th in those water should be lowered by the sinking of those riverine particles. Once the water was transferred to the region of sampling, it should represent an integrated signal mostly derived from flooding particle export not just the local marine particle export. Then even we have carefully estimated the horizonal and vertical transport of 234Th, the final 234Th flux is still not
induced by the local export. I do not know for this case 234Th is still a good tracer or not?

**Response: Please refer to our previous response to this comment in our last revision. We had agreed that flooding is not likely the main source of additional U to our study site. The discussion on U input via flooding had been removed in our last revision, so that the**
**reviewer's comment is not relevant to the current version of the manuscript.**

(Mark-up version: Changes made in the revision were marked with an underline.)

[revised manuscript text omitted]

Figure 2

[Figure]

Figure 3

Figure 4

[Figure]

Figure 5

[Figure]

[Figure]

Figure 6

Figure 7